# Learning in Structured Stackelberg Games

**Maria-Florina Balcan** [* 1]  **Kiriaki Fragkia** [* 1]  **Keegan Harris** [* 2]

## Abstract

We initiate the study of *structured Stackelberg games*, a novel form of strategic interaction between a leader and a follower where contextual information can be predictive of the follower's (unknown) type. Motivated by applications such as security games and AI safety, we show how this additional structure can help the leader learn a utility-maximizing policy in both the online and distributional settings. In the online setting, we first prove that standard learning-theoretic measures of complexity do not characterize the difficulty of the leader's learning task. We find that there exists a learning-theoretic measure of complexity, analogous to the Littlestone dimension in online classification, that *tightly* characterizes the leader's instance-optimal regret. We term this the *Stackelberg-Littlestone dimension*, and leverage it to provide a provably optimal online learning algorithm. In the distributional setting, we provide analogous results by showing that two new dimensions control the sample complexity upper- and lower-bound.

## 1. Introduction

Stackelberg games are the canonical framework for analyzing strategic interactions with commitment, where one player (the leader) commits to a strategy before another (the follower) observes this strategy and best-responds. While initially introduced in the context of market competition (von Stackelberg, 1934), Stackelberg games have been applied across a wide range of domains including security, where a security agency commits to a patrol strategy before attackers choose vulnerable targets to attack (Sinha et al., 2018), congestion control, where a city sets tolls before drivers choose routes (Swamy, 2012), and AI red-teaming, where AI providers deploy a model before users find adversarial use cases (Shivadekar, 2025; Liu et al., 2025).

*Contextual Stackelberg games* (Harris et al., 2024; Balcan et al., 2026) are a generalization of this model, in which both players' payoffs depend on additional side information. This model is motivated by many applications where players have payoff-relevant information about the state of the world at the time of play, which influences their strategies (and subsequently the outcome of the game). Side information may take the form of a feature vector and/or an embedding of natural language. For example, a wildlife park's security might place varying levels of importance on patrolling different areas of the park depending on the time of year, animal migration patterns, and recent reports describing suspicious activity or poaching risk. Contextual Stackelberg games have also been used to do label-efficient fine-tuning of large language models (Wang et al., 2026b) and reward shaping for inference–time alignment (Wang et al., 2026a).

Interestingly, learning under this more general framework is not always possible. Specifically, Harris et al. (2024) study the *online* setting (where the leader faces a sequence of followers with unknown utilities) and provide no-regret learning guarantees for the leader as long as either the followers or contexts are drawn i.i.d. from a fixed distribution. However, when both contextual information and follower types are chosen adversarially, the worst-case regret can grow linearly with the time horizon. This lower bound follows from a reduction to online classification, where an adversary can encode a mapping from contexts to follower types that is arbitrarily difficult to learn online.

In this work, we ask how broadly this impossibility result applies, and we investigate the landscape of learnability in this setting. Specifically, we examine learnability in contextual Stackelberg games under an arbitrary, unknown mapping from contexts to follower types. This setting naturally captures domains where contextual data contains some signal that is predictive of the follower about to appear next. In wildlife protection for example, data from motion sensors and camera footage is routinely collected and used to predict the intrusion type (Kamminga et al., 2018). Similarly in AI safety, different types of attack may be more or less viable depending on the domain the model is deployed in.

---

[1]Carnegie Mellon University, Pittsburgh, USA. [2]University of California, Berkeley, USA. Correspondence to: Maria-Florina Balcan <ninamf@cs.cmu.edu>, Kiriaki Fragkia <kiriakif@cs.cmu.edu>, Keegan Harris <keegan.harris@berkeley.edu>.

*Proceedings of the 43ʳᵈ International Conference on Machine Learning*, Seoul, South Korea. PMLR 306, 2026. Copyright 2026 by the author(s).

We capture such applications via the *structured* Stackelberg game model, which we introduce.

We study the following fundamental learnability questions:

*(1) In which structured Stackelberg games can the leader learn a utility-maximizing policy?*
*(2) Are there natural dimension(s) that characterize learnability in this setting?*

## 1.1. Our contributions

**Instance-optimal regret bounds.** We initiate the study of online learning in contextual Stackelberg games under an arbitrary, unknown mapping from contexts to follower types. In particular, we study how the leader can learn a utility-maximizing policy, given a hypothesis class rich enough to include this underlying mapping.

Naturally, the difficulty of the leader's learning task scales with the complexity of the hypothesis class. In the learning theory literature, the (multiclass) Littlestone dimension (Littlestone, 1987; Daniely et al., 2015) is used to characterize the complexity of online classification in settings with a finite set of labels (which, in structured Stackelberg games, corresponds to the set of follower types). We prove that the Littlestone dimension can give arbitrarily loose regret bounds in our Stackelberg game setting (Theorem 3.5). Our construction leverages the fact that the Littlestone dimension is completely blind to the utility space defined by the Stackelberg game, which ultimately determines the performance of the leader.

We introduce a new notion of complexity, the Stackelberg-Littlestone dimension (Definition 3.7; henceforth SL dimension), designed to capture the joint complexity of the hypothesis class and the utility space of the Stackelberg game. Our definition is based on the notion of shattered trees, which are a classic tool in online learning. However in contrast to previous work, our setting involves a discrete set of follower types, yet a continuous utility function, so constructing the trees requires a more nuanced approach. We prove that the SL dimension tightly characterizes the complexity of learning in structured Stackelberg games. Specifically, we prove that for any hypothesis class and any Stackelberg game, the SL dimension both upper- and lower-bounds the optimal regret for the leader (Theorem 3.8, Theorem 3.9). The proof of our upper bound is constructive (Algorithm 1), which yields a complete characterization of learnability in online structured Stackelberg games.

**PAC learning guarantees.** We also formalize the study of *distributional* learning in structured Stackelberg games (Section 4). In this setting, the leader has access to a set of (context, follower type) examples, where the contexts are drawn from some fixed, unknown distribution and the follower types are determined by some hypothesis in the hypothesis class. Upon the draw of a new context, the leader's goal is to deploy a strategy such as to maximize her expected utility.

We show generalization guarantees with probably approximately correct (PAC)-style bounds. We introduce the notion of the $\gamma$-SN dimension (Definitions 4.2 and 4.3) and the $\gamma$-SG dimension (Definitions 4.5 and 4.6), which control the lower- and upper-bounds on the sample complexity (Theorems 4.4 and 4.7). Our definitions are inspired by analogous notions in classification and regression (Natarajan, 1989; Attias et al., 2023; Daniely & Shalev-Shwartz, 2014), but we rely on a tailored approach to what constitutes a "mistake" in our setting. Our approach leverages the discreteness of the set of follower types to determine how much hypotheses differ in a minmax sense. We give a simple, improper algorithm (Algorithm 2) that with high probability achieves nearly-optimal expected utility with as many samples as in our sample complexity upper bound.

In the Appendix, we analyze a simpler, context-free offline setting, in which the training set is only over the distribution of follower types. Using tools from data-driven algorithm design, we show strong generalization guarantees (Theorem C.8). Moreover, all of our results directly extend to the related problems of learning to bid in auctions with side information and Bayesian persuasion with public and private states. We discuss this in Appendix A. Finally, computational considerations are discussed in Section 5.

## 1.2. Related work

**Learning in Stackelberg games.** The problem of learning an optimal strategy for the leader with a single unknown follower type has been studied by Letchford et al. (2009) and Blum et al. (2014). These results have been extended by Peng et al. (2019), who give faster rates and almost-matching lower bounds. Recently, Bacchiocchi et al. (2025) relax some structural assumptions from previous work, including tie-breaking rules for the follower, and revise worst-case sample complexity results of learning the leader's optimal mixed strategy.

Online Stackelberg games with multiple follower types were introduced in Balcan et al. (2015) and extended to online meta-learning in Harris et al. (2023). The authors show that when a series of unknown attacker types is chosen adversarially, playing the Hedge algorithm (Freund & Schapire, 1997) over a carefully selected set of mixed strategies is sufficient to obtain no-regret.

Our departing point is the work of Harris et al. (2024), in which the authors study the problem of online learning in *Stackelberg games with side information*. The authors show that when either the sequence of follower types or the sequence of contexts are drawn independently from some

fixed, unknown distribution, no-regret learning is indeed possible. Follow-up work (Balcan et al., 2026) obtains better regret guarantees under bandit feedback. However, when both sequences are chosen adversarially, regret can scale linearly in the worst case, as the problem can be reduced from online learning linear thresholds. We circumvent this impossibility result by making an additional structural assumption about the relationship between contexts and follower types—namely that it belongs to a class of functions that the leader has access to. Under this assumption, we show that online learning is indeed possible, and our guarantees depend on an appropriate measure of complexity of this function class.

Wang et al. (2026b) cast the problem of budget-aware supervised fine-tuning of large language models as a contextual Stackelberg game, and provide results for the setting where the leader does not observe the follower's type by default, but can query it at a cost.

**Multiclass learnability.** Our work can also be viewed through the lens of *multiclass learnability*, as it suffices for the leader to (1) learn the hypothesis class of the function mapping the contexts to a (finite) set of follower types, then (2) compute the optimal strategy against the predicted follower. Multiclass prediction has been studied extensively both in the distributional ((Natarajan, 1989; Daniely et al., 2015; Brukhim et al., 2022)) and online ((Daniely et al., 2015; Hanneke et al., 2023)) settings. Our results show that just relying on multiclass prediction tools that do not leverage the additional structure of the problem can lead to loose regret guarantees. For example, it is possible to construct a problem instance in which the leader can learn to play optimally under a hypothesis class that has infinite Littlestone dimension.

**Learning theory beyond multiclass prediction.** In online realizable regression Daskalakis & Golowich (2022) designed a proper algorithm that achieved near-optimal cumulative absolute loss with respect to the sequential fat-shattering dimension of the hypothesis class. Our work is more closely related to Attias et al. (2023), who introduce the online (resp. $\gamma$-Graph) dimension of a function class and show that it tightly characterizes online regression (resp. distributional learning) in the realizable setting up to a factor of 2. While we also consider utility functions with continuous image sets, the dimensions we introduce explicitly exploit the discrete structure of the Stackelberg game that arises from the finite set of follower types. Ahmadi et al. (2024) introduces a new combinatorial measure, the Strategic Littlestone Dimension, to achieve instance optimal mistake bounds in the online strategic classification setting. While both their measures of complexity and ours characterize learning in various principal-agent settings with commitment, the two are generally incomparable due to the differences in our settings.

**Learning piecewise Lipschitz functions.** Given the piecewise linear structure of the leader's utility function, our work also connects to the line of work on learning functions that are piecewise Lipschitz (Balcan et al., 2018a; Sharma et al., 2020; Dick et al., 2020; Balcan et al., 2021; 2018b). Specifically, we draw from this literature in the part of our work that explores a simpler distributional setting without contexts (Appendix C.1). In particular, the well-structured payoff function of the leader against a fixed follower type allows us to adopt tools from Balcan et al. (2018b) and get better generalization guarantees compared to prior work that studies a similar setting (Letchford et al., 2009).

## 2. Setting and notation

We use $[N] := \{1, 2, \ldots, N\}$ to denote the set of natural numbers up to and including $N$ and $\Delta(\mathcal{A})$ to denote the probability simplex over a (finite) set $\mathcal{A}$. For strings, we use inclusive slice indexing such that $x[i:j]$ denotes the entries of $x$ from position $i$ to $j$. Dropping either $i$ or $j$ means that the slice extends from the beginning of the string or to the end, respectively. We also use negative indexing, so that, e.g., $x[-1]$ is the last element of $x$. All proofs are deferred to the Appendix.

We consider a Stackelberg game between a leader and a follower. Before the game starts, some contextual information $\mathbf{z} \in \mathcal{Z} \subseteq \mathbb{R}^d$ is revealed to both players. The leader plays first and commits to a mixed strategy $\mathbf{x} \in \Delta(\mathcal{A})$, where $\mathcal{A}$ is the set of finite actions available to the leader[1]. The follower *best-responds* to the leader's strategy and the context by playing some action $a_f \in \mathcal{A}_f$, where $\mathcal{A}_f$ is the finite set of follower actions.[2] The follower's best-response is defined as

$$b_f(\mathbf{z}, \mathbf{x}) \in \arg\max_{a_f \in \mathcal{A}_f} \sum_{a_l \in \mathcal{A}} \mathbf{x}[a_l] \cdot u_f(\mathbf{z}, a_l, a_f),$$

where, $u_f : \mathcal{Z} \times \mathcal{A} \times \mathcal{A}_f \to [0, 1]$ is the follower's utility function. In case of ties, we assume that the follower uses a fixed, known, and deterministic tie-breaking order over $\mathcal{A}_f$.

We allow the follower to be one of $K$ follower types in the set $\{f^{(1)}, \ldots, f^{(K)}\}$. The type of follower characterizes their utility function, $u_{f^{(i)}} : \mathcal{Z} \times \mathcal{A} \times \mathcal{A}_f \to [0, 1]$. We assume that the leader knows the set of possible follower types *a priori*[3], but not the realized type and aims to maximize her own utility, $u : \mathcal{Z} \times \mathcal{A} \times \mathcal{A}_f \to [0, 1]$. By slight abuse of notation, we sometimes use the shorthand $u(\mathbf{z}, \mathbf{x}, a_f) := \sum_{a_l \in \mathcal{A}} \mathbf{x}[a_l] \cdot u(\mathbf{z}, a_l, a_f)$.

We will denote a Stackelberg game as a tuple, $\mathcal{G} =$

---

[1]Our results hold for any strategy set that the leader can optimize over.

[2]WLOG the follower's best-response is a pure strategy.

[3]This assumption is for ease of exposition. Our results depend only on the leader knowing the best response function for each follower type.

$(\mathcal{A}, \mathcal{A}_f, \mathcal{Z}, u, u_{f^{(1)}}, \ldots, u_{f^{(K)}})$, consisting of the leader action set, the follower action set, the set of possible contexts, as well as the leader and followers' utility functions.

*Remark* 2.1. For the remainder of the paper, and without loss of generality, we assume that the follower types in $[K]$ are *distinct*. We say that two follower types are distinct if there exists some context for which there is no single leader strategy is simultaneously optimal against both types.

Let $\mathcal{H} \subseteq [K]^{\mathcal{Z}}$ be a set of candidate functions (also called a *hypothesis class*) mapping contexts to follower types, which the leader uses to learn. Throughout, we focus on the *realizable setting*, i.e., we assume that $\mathcal{H}$ is sufficiently rich to contain a perfect mapping $h^* \in \mathcal{H}$, where $h^*(\mathbf{z}_t) = f_t$ for all $t \in [T]$. We use $h^*$ as a benchmark for the leader's performance. For a single interaction with a follower at context $\mathbf{z}$, we write the leader's loss when playing strategy $\hat{\mathbf{x}} \in \Delta(\mathcal{A})$ as

$$r(\mathbf{z}, \hat{\mathbf{x}}, f^{(h^*(\mathbf{z}))}) = \sup_{\mathbf{x} \in \Delta(\mathcal{A})} u\left(\mathbf{z}, \mathbf{x}, b_{f^{(h^*(\mathbf{z}))}}(\mathbf{z}, \mathbf{x})\right)$$
$$- u\left(\mathbf{z}, \hat{\mathbf{x}}, b_{f^{(h^*(\mathbf{z}))}}(\mathbf{z}, \hat{\mathbf{x}})\right).$$

We let $\pi_{h^*}(\mathbf{z}) \in \arg\sup_{\mathbf{x} \in \Delta(\mathcal{A})} u(\mathbf{z}, \mathbf{x}, b_{h^*(\mathbf{z})}(\mathbf{z}, \mathbf{x}))$ denote the optimal policy for the leader.

For a hypothesis class $\mathcal{H}$, we denote its projection on a point as $\mathcal{H}(\mathbf{z}) = \{i \in [K] : \exists h \in \mathcal{H} \text{ s.t. } h(\mathbf{z}) = i\}$ and on a set as $\mathcal{H}(\{\mathbf{z}_i\}_{i \in [n]}) = \{(h(\mathbf{z}_1), \ldots, h(\mathbf{z}_n)) : h \in \mathcal{H}\}$. For $i \in [K]$, we write $\mathcal{H}^{(\mathbf{z} \to i)} = \{h \in \mathcal{H} : h(\mathbf{z}) = i\}$.

## 3. Online learning

In the online setting, the leader is faced with a sequence of followers. In each round $t \in [T]$, a context $\mathbf{z}_t \in \mathcal{Z}$ is revealed to both players. A follower of type $f_t$ arrives, the leader commits to a mixed strategy $\mathbf{x}_t \in \Delta(\mathcal{A})$, and the follower best-responds. We assume that the leader knows the utilities of all follower types, but the type of the follower at time $t$ is not revealed to the leader until after the round is over. We make no assumptions about how the sequence of contexts is chosen, i.e., they may be chosen adversarially.

Next we present the notion of *contextual Stackelberg regret*, which quantifies the leader's performance over time.

**Definition 3.1** (Contextual Stackelberg Regret). Given a sequence of pairs of contexts and followers $\{(\mathbf{z}_t, f_t)\}_{t=1}^{T}$ the leader's regret is $R(T) := \sum_{t=1}^{T} r(\mathbf{z}_t, \mathbf{x}_t, f_t)$ where $\mathbf{x}_t \in \Delta(\mathcal{A})$ is the leader's strategy at timestep $t$.

The goal of the leader is to minimize her regret, that is, to achieve utility comparable to the utility she would have obtained if she could correctly predict the follower's type given the context and play optimally against it.

*Remark* 3.2. Our analysis relies on the leader being able to compute the optimal strategy to play for any fixed context

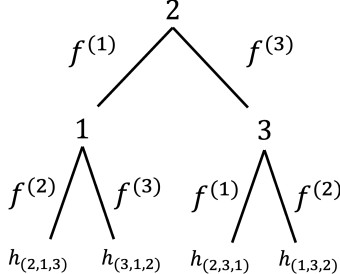

*Figure 1.* $\mathcal{H}_3$-shattered Littlestone Tree showing the consistent hypothesis for each root-to-leaf path. $h_{(a,b,c)}$ denotes the hypothesis with $h(1) = f^{(a)}$, $h(2) = f^{(b)}$, and $h(3) = f^{(c)}$.

and follower type. However, due to the tie-breaking rules of the follower, the leader may not be able to get an exact solution. Previous work (Balcan et al., 2015; Harris et al., 2024) shows that the leader can get an arbitrarily small error in the utility by optimizing over an appropriately selected subset of the simplex. For ease of exposition, we assume that the leader can directly optimize over the entire simplex, without affecting her regret guarantees. A formal statement of this result appears in Appendix B.1.

### 3.1. Littlestone dimension

Given the structure induced by the finite set of follower types, a natural idea is to directly predict the follower's type from the context, and play the corresponding optimal strategy. This suggests using tools from online (multiclass) classification to characterize the statistical complexity of the problem. In this section, we will show that this approach comes up short.

We begin by recalling several concepts from online learning that are relevant to our work, starting with the notion of *shattering*, which captures the intuition that richer hypotheses classes can encode more complex label patterns.

**Definition 3.3** (Multiclass Littlestone Tree). A Littlestone tree is a rooted tree, in which each internal node is labeled with an instance in $\mathcal{Z}$ and each edge corresponds to a label in a finite set $[K]$. We say that the tree is *shattered* by a hypothesis class $\mathcal{H}$ if for every root-to-leaf path that traverses nodes $\mathbf{z}_1, \ldots, \mathbf{z}_n$, there exists a hypothesis $h \in \mathcal{H}$ such that the label of the edge in the path leaving $\mathbf{z}_i$ is $h(\mathbf{z}_i)$ for all $i$.

**Example 1.** *Consider the class of permutations* $\mathcal{H}_3 = \{h : \{1, 2, 3\} \to \{f^{(1)}, f^{(2)}, f^{(3)}\} \mid h \text{ is a bijection}\}$. *Figure 1 shows an example of a Littlestone Tree that is shattered by this class.*

The *multiclass Littlestone dimension* reflects how the ability of a function class to shatter deeper trees corresponds to greater complexity in the class.[4] This definition

---

[4]For binary labels and when all nodes at the same depth of a

is due to Daniely et al. (2015) and it generalizes the classic definition of the Littlestone dimension for binary classification (Littlestone, 1987).

**Definition 3.4** (Multiclass Littlestone Dimension)**.** The Littlestone dimension of a multiclass hypothesis class $\mathcal{H}$, denoted Ldim($\mathcal{H}$), is the maximal integer $d$ such that there exists a full binary tree of depth $d$ that is shattered by $\mathcal{H}$.

The finiteness of the Littlestone dimension of a hypothesis class is necessary for a hypothesis class to be online learnable (Littlestone, 1987). Some examples of classes with finite Littlestone dimension are conjunctions, decision lists, linear separators with a margin, and in general anything learnable in the mistake bound model (cf. Balcan (2011), Proposition 17 in Alon et al. (2021)).

Daniely et al. (2015) show that the multiclass Littlestone dimension of a hypothesis class characterizes learnability in online multiclass classification, that is, it determines the number of mistakes an optimal online learner will make in the worst-case. In our setting, learning how to optimally predict the follower's type given a context is an instance of multiclass prediction and is indeed *sufficient* for learning an optimal policy. Namely, given the context and the predicted follower type, the leader can compute the strategy that maximizes her utility by solving a sequence of $|\mathcal{A}_f|$ linear programs (Conitzer & Sandholm, 2006).

However, it is *a priori* unclear whether learning to optimally predict the follower's type is not only sufficient but also necessary. We now show that this is *not* the case, by constructing a structured Stackelberg game instance with an infinite Littlestone dimension that can be solved without learning.

**Theorem 3.5.** *There exists a Stackelberg game $\mathcal{G}$ and a hypothesis class $\mathcal{H}$ such that* Ldim($\mathcal{H}$) $= \infty$*, but the leader's optimal policy can be determined without learning.*

*Proof sketch.* The key idea is that, although the follower types can be distinct from the leader's perspective (Remark 2.1) and the hypothesis class can encode arbitrarily complex type-prediction problems, the leader's downstream utility optimization problem may be much simpler. Our construction uses two follower types and a hypothesis class, $\mathcal{H}$, consisting of linear threshold functions. Learning the true threshold can be hard (Ldim($\mathcal{H}$) $= \infty$), but is unnecessary: near the threshold the two follower types induce the same optimal strategy for the leader. Figure 2 provides an illustration of the Stackelberg game we use in the proof. $\square$

This result shows that a finite Littlestone dimension is not necessary for online learning in our setting, as it only

shattered tree are the same, the Littlestone dimension recovers the definition of the well-studied VC-dimension.

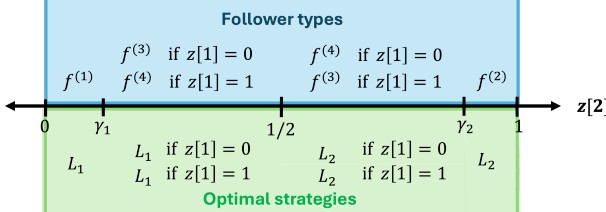

*Figure 2.* Follower types as a function of the context in the construction of Theorem 3.5. While $\gamma_1$ and $\gamma_2$ vary with each hypothesis, the leader's optimal strategy only depends on whether $\mathbf{z}[2] < 0.5$.

measures the complexity of predicting the follower types, not how those predictions affect the leader's utilities.

### 3.2. Stackelberg Littlestone dimension

We now define the Stackeleberg Littlestone dimension, which we prove characterizes online learnability in structured Stackeleberg games.

In the definitions that follow we consider trees with nodes labeled by elements in $\mathcal{Z}$ and edges corresponding to different labels in $[K]$. A depth-$d$ tree, $\mathcal{T}_d = \{\mathbf{z}_s : s \in S_d\}$, is indexed by sequences $S_d \subseteq \{\varnothing\} \cup \{s \in [K]^\ell : 1 \le \ell \le d\}$, where $\mathbf{z}_\varnothing$ denotes the root node. We use $z_{s \le d}$ to denote a context $z_s$ with depth at most $d$.

**Definition 3.6** (Stackelberg Littlestone (SL) Tree)**.** A SL tree, $\mathcal{T}_d$, is a rooted tree, in which each internal node is labeled with an instance in $\mathcal{Z}$ and each edge corresponds to a label in $[K]$. Each node $\mathbf{z}_s \in \mathcal{T}_d$ has a weight $\rho_s$, where $\rho_s = 0$ if $\mathbf{z}_s$ is a leaf node and $\rho_s = \inf_{\mathbf{x} \in \Delta(\mathcal{A})} \max_{j \in [K]:sj \in S_d} \left( r(\mathbf{z}_s, \mathbf{x}, f^{(j)}) + \rho_{sj} \right)$ otherwise. We say that the tree is *shattered* by a hypothesis class $\mathcal{H}$ under Stackelberg game $\mathcal{G}$ if for every root-to-leaf path that traverses nodes $\mathbf{z}_\varnothing, \ldots \mathbf{z}_{s \le d}$, there exists $h \in \mathcal{H}$ such that the label of the edge in the path leaving $\mathbf{z}_{s \le i}$ is $h(\mathbf{z}_{s \le i})$ for every $i$.[5]

**Example 2.** *Consider the hypothesis class of permutations*

$$\mathcal{H}_3 = \left\{ h : \{1, 2, 3\} \mapsto \{f^{(1)}, f^{(2)}, f^{(3)}\} \Big| h \text{ is a bijection} \right\}$$

*and a Stackelberg game with 3 actions for both players. The leader's utility matrix is $U_L = \mathbb{I}_{3 \times 3}$ and follower $f^{(i)}$'s utility matrix is $U_{f^{(i)}} = \mathbf{1} e_i^\top$, i.e. she receives utility 1 if and only if she plays action $i$. Figure 3 shows a corresponding SL tree.*

**Definition 3.7** (Stackelberg Littlestone (SL) dimension)**.** The SL dimension of a hypothesis class $\mathcal{H}$ for Stackelberg game $\mathcal{G}$ is defined as SLdim$_{\mathcal{G}}(\mathcal{H}) := \sup\{\kappa \ge 0 :$ there exists an SL tree shattered by $\mathcal{H}$ under $\mathcal{G}$ with root node weight $\kappa\}$.

[5]$z_{s \le d}$ refers to a context $z_s$ with depth at most $d$.

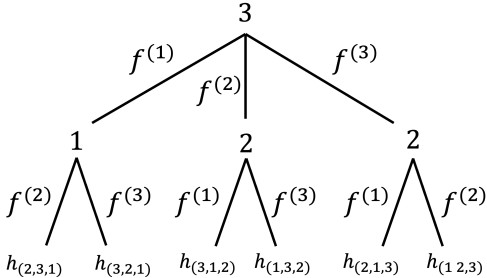

*Figure 3.* $\mathcal{H}_3$-shattered SL-Tree showing a consistent hypothesis for each root-to-leaf path. Leaf nodes have weight 0, nodes at depth 1 have weight $1/2$ and the root node has weight $1/2 + 2/3$.

Note that taking the supremum is necessary since it is possible that no finite-depth tree achieves the optimal value. If for every $\kappa \geq 0$ there exists an SL tree with root node weight at least $\kappa$, then $\mathrm{SLdim}_\mathcal{G}(\mathcal{H}) = \infty$.

We now show that no deterministic algorithm can achieve regret lower than the SL dimension.

**Theorem 3.8.** *For any hypothesis class $\mathcal{H}$ and Stackelberg game $\mathcal{G}$, there exists an adversarial sequence of contexts and follower types such that any deterministic algorithm suffers at least $\mathrm{SLdim}_\mathcal{G}(\mathcal{H}) - \epsilon$ regret, for any $\epsilon > 0$.*

*Proof sketch.* The key idea in the proof is that the weight assigned to each node of an SL tree measures how much regret the adversary can still guarantee from that point onward. At the root, this weight captures the minimum regret the learner must suffer no matter which strategy she plays. If the learner chooses the strategy that minimizes immediate regret, the adversary selects the follower type that leads into a subtree with large remaining weight, ensuring future regret. If the learner deviates to reduce future regret, the adversary instead extracts more regret immediately.

By repeating this reasoning along a root-to-leaf path, the adversary can force the learner to incur regret equal to the root weight of the tree. Since the SL dimension is defined as the supremum root weight over all such shattered trees, no deterministic algorithm can achieve regret smaller than the SL dimension (up to an arbitrarily small additive term). $\square$

We now present an instance-optimal algorithm that achieves regret equal to the SL dimension. We call Algorithm 1 the Stackelberg Standard Optimal Algorithm (SSOA), as it shares some similarities with the Standard Optimal Algorithm (SOA) from online multiclass prediction (Daniely & Shalev-Shwartz, 2014). In each round, SSOA tracks the subset of hypotheses that are consistent with the history so far. The algorithm then plays the mixed strategy that minimizes the worst-case instantaneous regret, plus the SL dimension of the induced hypothesis class. The latter

captures the difficulty of the learning task that remains after the current round ends.

---

**Algorithm 1** SSOA

---

1: **Input:** Stackelberg game $\mathcal{G}$, hypothesis class $\mathcal{H}$
2: Initialize version space $V_1 = \mathcal{H}$
3: **for** $t \in [T]$ **do**
4:     Observe context $\mathbf{z}_t$
5:     Compute optimal utility for each type $i \in V_t(\mathbf{z}_t)$
     $u_*^{(i)} = \sup_{\mathbf{x} \in \Delta(\mathcal{A})} u(\mathbf{z}_t, \mathbf{x}, b_{f^{(i)}}(\mathbf{z}_t, \mathbf{x}))$
6:     Play $\mathbf{x}_t \in \arg\inf_{\mathbf{x} \in \Delta(\mathcal{A})} \max_{j \in V_t(\mathbf{z}_t)} \left( u_*^{(j, \mathbf{z}_t)} - u(\mathbf{z}_t, \mathbf{x}, b_{f^{(j)}}(\mathbf{z}_t, \mathbf{x})) + \mathrm{SLdim}_\mathcal{G}\left(V_t^{(\mathbf{z}_t \to j)}\right) \right)$
7:     Observe follower type $f_t$
8:     Update version space $V_{t+1} = V_t^{(\mathbf{z}_t \to f_t)}$
9: **end for**

---

**Theorem 3.9.** *For any Stackelberg game $\mathcal{G}$ and every hypothesis class $\mathcal{H}$, the regret of Algorithm 1 is at most $\mathrm{SLdim}_\mathcal{G}(\mathcal{H})$.*

*Proof sketch.* Our proof follows a high-level idea similar to the analysis of SOA in online prediction: With each mistake, the learner's task becomes easier, as the Littlestone dimension decreases. Similarly, we show that when the learner incurs instantaneous regret, the SL dimension of the induced hypothesis class decreases by at least this amount. $\square$

### 3.3. Relationship with the Littlestone dimension

We have so far shown that the SL dimension characterizes learnability in structured Stackelberg games. We now examine how the it relates to online multiclass prediction.

**Lemma 3.10.** *For any hypothesis class $\mathcal{H}$ and any Stackelberg game $\mathcal{G}$, $\mathrm{SLdim}_\mathcal{G}(\mathcal{H}) \leq \mathrm{Ldim}(\mathcal{H})$.*

To make the relationship between the two dimensions more concrete, consider the following example, which is a generalization of Examples 1 and 2 to $n$ dimensions.

**Example 3.** *Consider a setting with $n$ contexts, $n$ follower types, and the class of permutations $\mathcal{H}_n$. Note that $\mathrm{Ldim}(\mathcal{H}_n) = n - 1$, since an adversary can force a mistake in every round that she can query. A tree of depth $n$ cannot be shattered because once the labels of the first $n - 1$ contexts have been observed, the label of the last context is fixed.*

*Now consider a Stackelberg game where the leader's utility matrix is $U = \mathbf{I}_{n \times n}$ and follower type $i$'s utility matrix is $U_i = \mathbf{1}_n e_i^\top$. We will show that $\mathrm{SLdim}_\mathcal{G}(\mathcal{H}_n) = n - H_n$, where $H_n$ is the $n$-th harmonic number.*

*Consider the SL tree that witnesses the SL dimension. Due to symmetry, each node at depth $d$ will have $n - d$ outgoing edges, one for each follower type not encountered in the*

*path so far. Each node at depth $n-2$ will have weight $1/2$, since the optimal utility of the leader against any follower type is $1$, the optimal utility against any follower chosen adversarially from a set of two types is $1/2$, and any deeper nodes must have only one outgoing edge (and therefore weight $0$). More generally, for a node $s$ at depth $d$ we have*

$$\rho_s = \inf_{\mathbf{x} \in \Delta(\mathcal{A})} \max_{j \in \{j_1, \dots, j_{n-d}\}} \left[ r(\mathbf{z}, \mathbf{x}, f^{(j)}) + \sum_{j=2}^{n-d-1} 1 - \frac{1}{j} \right],$$

*which simplifies to $n - d - \sum_{j=1}^{n-d} \frac{1}{j}$.*

Note that while any deterministic learner will make at least $\mathrm{Ldim}(\mathcal{H})$ mispredictions when learning $\mathcal{H}$, mispredicting the follower's type need not lead to suboptimal utility for the leader. Therefore, Lemma 3.10 does not immediately imply that the optimal multiclass prediction algorithm (SOA) would incur suboptimal regret in our setting. Indeed, in Theorem 3.5 we constructed a game instance where the leader can mispredict the follower's type at every timestep, yet still achieve optimal utility. We conclude this section by showing that there are indeed game instances where SOA will suffer larger regret than SSOA.

**Theorem 3.11.** *There exists a hypothesis class $\mathcal{H}$ and Stackelberg game $\mathcal{G}$ such that running SOA on $\mathcal{H}$ results in suboptimal utility for the leader.*

*Proof sketch.* The game construction exploits the mismatch between the label prediction objective of SOA and optimizing for leader utility. Specifically, the game is designed such that SOA focuses on parts of the hypothesis class that are combinatorially complex (large Littlestone dimension) but strategically irrelevant. As a result, SOA unnecessarily suffers utility loss over multiple rounds. $\square$

# 4. Distributional learning

We now shift our focus from online learning to distributional learning. In this setting, contexts are drawn i.i.d. from an unknown distribution and as in the online setting, we assume that there exists a function $h^* \in \mathcal{H}$ that correctly predicts each follower type given the context. The leader's goal is to choose a strategy that will maximize her expected utility on a fresh sample drawn from the same distribution. A learning algorithm is a mapping $A : (\mathcal{Z} \times [K])^* \to \Delta(\mathcal{A})^{\mathcal{Z}}$.

We consider the following notion of sample complexity, adapted to our Stackelberg setting.

**Definition 4.1** (PAC cut-off sample complexity). The PAC cut-off sample complexity $m(\mathcal{H}, \mathcal{G}; \epsilon, \delta, \gamma)$ of function class $\mathcal{H}$ w.r.t. Stackelberg game $\mathcal{G}$ is $m(\mathcal{H}, \mathcal{G}; \epsilon, \delta, \gamma) = \inf_A m_A(\mathcal{H}, \mathcal{G}; \epsilon, \delta, \gamma)$, where $m_A(\mathcal{H}, \mathcal{G}; \epsilon, \delta, \gamma)$ is the smallest integer such that for any $m \geq m_A(\mathcal{H}, \mathcal{G}; \epsilon, \delta, \gamma)$, every distribution $\mathcal{D}$ on $\mathcal{Z}$, and any target hypothesis

$h^* \in \mathcal{H}$, the expected cut-off loss of algorithm $A$, $\mathbb{P}_{\mathbf{z} \sim \mathcal{D}}[r(\mathbf{z}, A(\mathbf{z}; S), f^{(h^*(\mathbf{z}))}) > \gamma]$, is at most $\epsilon$ with probability at least $1 - \delta$.

While we present our results with respect to the PAC cut-off sample complexity, any bound with respect to this benchmark immediately implies a bound on the familiar PAC sample complexity (Valiant, 1984) as well. [6]

## 4.1. Sample complexity lower bound

We now introduce the $\gamma$-valued Stackelberg-Natarajan dimension (henceforth SN dimension), which we show characterizes the sample complexity lower bound for distributional learning in structured Stackelberg games. Indeed, we prove that the finiteness of this measure is a necessary condition for learnability in our setting. The SN dimension is inspired by the Natarajan dimension from multiclass prediction (Natarajan, 1989; Daniely & Shalev-Shwartz, 2014) and the $\gamma$-Natarajan dimension from realizable regression (Attias et al., 2023).

We start by defining what it means for a set of contexts to be $\gamma$-SN-shattered. Intuitively, a set is $\gamma$-SN-shattered if the hypothesis class $\mathcal{H}$ can fit complex patterns of follower types on the set. Furthermore, these follower types must be sufficiently different: no leader strategy can incur loss less than $\gamma$ if the type is chosen adversarially among them.

**Definition 4.2** ($\gamma$-SN-shattered set). A set $\{\mathbf{z}_1, \dots, \mathbf{z}_n\} \subseteq \mathcal{Z}$ is $\gamma$-SN-shattered by hypothesis class $\mathcal{H}$ with respect to Stackelberg game $\mathcal{G}$ if there exist functions $g_0, g_1 : \mathcal{Z} \to [K]$ such that

1. $\inf_{\mathbf{x} \in \Delta(\mathcal{A})} \max_{j \in \{g_0(\mathbf{z}_i), g_1(\mathbf{z}_i)\}} r(\mathbf{z}_i, \mathbf{x}, f^{(j)}) > \gamma$ for every $\mathbf{z}_i \in S$, and
2. for all $b \in \{0, 1\}^n$, there exists an $h_b \in \mathcal{H}$ such that

$$h_b(\mathbf{z}_i) = \begin{cases} g_0(\mathbf{z}_i) & \text{if } b_i = 0 \\ g_1(\mathbf{z}_i) & \text{if } b_i = 1. \end{cases}$$

**Definition 4.3** ($\gamma$-valued SN dimension). The $\gamma$-valued SN dimension, $\mathrm{SNdim}_{\mathcal{G}}^{(\gamma)}(\mathcal{H})$, is the cardinality of the largest set that is $\gamma$-SN-shattered by $\mathcal{H}$ with respect to $\mathcal{G}$.

We emphasize that that the two functions, $g_0$ and $g_1$ in Definition 4.2 must be $\gamma$-far from each other in a minmax sense: at each context the leader cannot choose a mixed strategy that performs well against both $g_0(\mathbf{z}_0)$ and $g_1(\mathbf{z}_0)$. This enables a distribution-independent lower bound that holds for every learning algorithm. We prove our lower bound in Theorem 4.4 by adapting the argument of Attias et al. (2023) to our setting.

**Theorem 4.4.** *Let $A$ be any learning algorithm and $\epsilon, \delta, \gamma \in$*

---

[6]See Lemma 1 in Attias et al. (2023).

$(0, 1)$ *such that* $\delta < \epsilon$, *then*

$$m_A(\mathcal{H}, \mathcal{G}; \epsilon, \delta, \gamma) \geq \Omega \left( \frac{\text{SNdim}_{\mathcal{G}}^{(\gamma)}(\mathcal{H}) + \log(1/\delta)}{\epsilon} \right).$$

*Proof sketch.* The high-level idea is to reduce distributional learning in structured Stackelberg games to a hard identification problem over a $\gamma$-SN-shattered set. Such a set contains many contexts where the hypothesis class can realize two different follower types such that no leader strategy can simultaneously perform well against both.

The distribution we consider on this shattered set has most probability mass on one "easy" context and small but non-negligible mass spread over many "hard" contexts. With limited samples, the learner is unlikely to observe enough of the hard contexts to correctly distinguish the follower type. Since the hypothesis class can label these unseen contexts arbitrarily while remaining consistent with the data, the learner's strategy on them is effectively a guess.

Because each unseen context independently forces a $\gamma$-loss with constant probability, the learner's expected cut-off error remains large unless the number of samples scales linearly with the size of the $\gamma$-SN-shattered set. Standard concentration arguments then show that achieving small error with high probability requires a sample size proportional to the $\gamma$-SN dimension. $\qquad\square$

## 4.2. Sample complexity upper bound

We conclude our treatment of the distributional setting by introducing the $\gamma$-valued Stackelberg-Graph dimension (henceforth SG dimension), which provides a sufficient condition for distributional learning in structured Stackelberg games. The SG dimension is a natural relaxation of the $\gamma$-Natarajan dimension and an adaptation of similar dimensions from classification (Natarajan, 1989; Daniely & Shalev-Shwartz, 2014) and regression (Attias et al., 2023) to our setting.

**Definition 4.5** ($\gamma$-SG-shattered set). A set $\{\mathbf{z}_1, \ldots, \mathbf{z}_n\} \subseteq \mathcal{Z}$ is $\gamma$-SG-shattered by hypothesis class $\mathcal{H}$ with respect to Stackelberg game $\mathcal{G}$ if there exists a function $g : \mathcal{Z} \to [K]$ such that for every $b \in \{0, 1\}^n$, there exists an $h_b \in \mathcal{H}$ satisfying

1. $h_b(\mathbf{z}_i) = g(\mathbf{z}_i) \ \forall i \in [n]$ such that $b_i = 0$, and
2. $\inf_{\mathbf{x} \in \Delta(\mathcal{A})} \max_{j \in \{g(\mathbf{z}_i), h_b(\mathbf{z}_i)\}} r(\mathbf{z}_i, \mathbf{x}, f^{(j)}) \geq \gamma$ for all $i \in [n]$ such that $b_i = 1$.

**Definition 4.6** ($\gamma$-valued SG dimension). The $\gamma$-valued SG dimension, $\text{SGdim}_{\mathcal{G}}^{(\gamma)}(\mathcal{H})$, is the cardinality of the largest set that is $\gamma$-SG-shattered by $\mathcal{H}$ with respect to $\mathcal{G}$.

We now present the main result of this section. Given a training sample $S$ of context–follower-type pairs, Algorithm 2

first computes the subset of hypotheses that are perfectly consistent with $S$. Then, on a new context, the algorithm plays the leader's strategy that is optimal against an adversarially chosen follower type predicted at that context by any hypothesis consistent with $S$.

---

**Algorithm 2** Stackelberg Distributional Learner ($\mathfrak{L}^*$)

1: **Input:** Stackelberg game $\mathcal{G}$, hypothesis class $\mathcal{H}$, sample $S = \{(\mathbf{z}_i, f_i)\}_{i \in [n]}$, and test context $\mathbf{z} \in \mathcal{Z}$
2: Compute $\mathcal{H}|_S = \{h \in \mathcal{H} : h(\mathbf{z}_i) = f_i \ \forall i \in [n]\}$
3: Compute $F = \mathcal{H}|_S(\mathbf{z}) = \{h(\mathbf{z}) \in [K] : h \in \mathcal{H}|_S\}$
4: Play $\mathbf{x}^* = \inf_{\mathbf{x} \in \Delta(\mathcal{A})} \max_{i \in F} r(\mathbf{z}, \mathbf{x}, f^{(i)})$

---

We now show that the $\gamma$-SG dimension upper bounds the PAC sample complexity in our setting. The proof of Theorem 4.7 follows from adapting that of Attias et al. (2023) to our setting.

**Theorem 4.7.** *For any class $\mathcal{H}$, game, $\mathcal{G}$, and $\epsilon, \delta, \gamma \in (0, 1)$, there is some constant $C_1$ such that Algorithm 2 achieves*

$$m_{\mathfrak{L}^*}^r(\mathcal{H}, \epsilon, \delta, \gamma) \leq C_1 \frac{\text{SGdim}_{\mathcal{G}}^{(\gamma)}(\mathcal{H}) \log(1/\epsilon) + \log(1/\delta)}{\epsilon}.$$

*Proof sketch.* The key observation in our analysis is that the leader only needs to worry about contexts where different hypotheses would force meaningfully different leader utilities. Algorithm 2 keeps all hypotheses that are consistent with the data and, given a new context, plays the strategy that is most robust to all follower types predicted by those hypotheses. If the learner performs poorly on a noticeable fraction of contexts, then there must exist many contexts where some hypothesis in the class disagrees with the true one and where this disagreement induces loss at least $\gamma$. Such contexts form a $\gamma$-SG-shattered set.

The definition of the $\gamma$-SG dimension limits the size of these hard sets. Using standard uniform convergence and symmetrization arguments, we show that once the sample size exceeds this dimension, no hypothesis consistent with the data can be $\gamma$-far from the true one on more than an $\epsilon$ fraction of contexts. As a result, the chosen strategy achieves near-optimal expected utility with high probability. $\qquad\square$

## 5. Computational complexity considerations

Computational complexity has been a longstanding concern in the analysis of Stackelberg games. In the setting where a leader interacts with a single (known) follower type, Conitzer & Sandholm (2006) show that computing the optimal mixed strategy for the leader can be achieved in polynomial time using linear programming. However in more complex Stackelberg games such as Bayesian Stackelberg games, Stackelberg security games, and even 3-player

Stackelberg games computing the leader's optimal strategy has been shown to be NP-hard (Conitzer & Sandholm, 2006; Korzhyk et al., 2010). Side information that is predictive of the follower's type may help us circumvent some of these impossibility results. We provide a preliminary analysis of computational complexity in our setting here, which results in polynomial runtimes and sublinear (although not necessarily optimal) regret guarantees.

In a structured Stackelberg game with hypothesis class $\mathcal{H}$, the leader's optimal policy can be learned in polynomial time if $\mathcal{H}$ is efficiently online learnable. We give a high-level overview of this approach. First, we consider a learning algorithm for the hypothesis class $\mathcal{H}$. The leader can use this algorithm to predict the follower's type and based on that compute the optimal mixed strategy against that type as if her prediction were correct. As shown by Conitzer & Sandholm (2006), for a fixed and known follower type, computing the leader's optimal mixed strategy can be achieved in polynomial time using linear programming. The learner will make a finite number of follower type mispredictions before finding $h^*$, and at this point will play optimally against every follower type that appears in every subsequent round. Therefore, for any hypothesis class in the realizable setting that is efficiently online- (and therefore also PAC-) learnable, we can establish a computationally efficient algorithm for learning the leader's optimal policy. We now present one such example of a hypothesis class that can be learned efficiently when used to predict the followers' types.

**Example 4.** *Consider the hypothesis class, $\mathcal{H}_{MDL}$ of multiclass decision lists. Each decision list is a sequence of "if-then" rules, specifically of the form "if $\ell_1$ then $k_1$, else if $\ell_2$ then $k_2$, else if $\ell_3$ then $k_3$, ..., else $k_{final}$". Each decision list $h \in \mathcal{H}$ can be represented by a list $[(\ell_1, k_1), \ldots, (\ell_n, k_n), (k_{final})]$ that consists of condition-value pairs $(\ell_i, k_i)$ and a final default value $(k_{final})$. Each condition $\ell_i$ is a literal (of either a variable in a set of input variables, $\mathcal{Z} = \{x_1, \ldots, x_n\}$, or its negation) and each value $k_i$ belongs to a set of output labels, $[K]$. A decision list is evaluated sequentially. It takes as input an assignment of variables in $\mathcal{Z}$ i.e. $\mathbf{z} \in \{0, 1\}^n$ and its output is the value corresponding to the first condition that is satisfied by the input, $\mathbf{z}$.*

**Theorem 5.1.** *The class of multiclass decision lists in Example 4 is efficiently online learnable.*

We present the proof of Theorem 5.1 in Appendix D.

## 6. Conclusion and future directions

We initiate the study of structured Stackelberg games, where contextual information is predictive of the follower's type. Our main results consist of a complete characterization of instance-optimal regret bounds in the online setting, and PAC sample complexity guarantees in the distributional setting. There are several exciting directions for future work.

**Computationally efficient algorithms.** Our main focus in this work is on the statistical complexity of learning in structured Stackelberg games. An interesting avenue for future research is to consider the computational complexity of learning in this setting. We conduct a preliminary exploration of this direction in in Section 5, and find that our results open the door for efficient (albeit so far sub-optimal) learning algorithms in some settings. It would be interesting to explore whether there are some settings where the structure introduced in our work can lead to both optimal and efficient algorithms, or if there is a fundamental tradeoff between computational complexity and tight regret guarantees.

**The agnostic setting.** We focus on the realizable setting, where the leader is given a class of hypotheses with the promise that it includes the optimal predictor. The agnostic learning setting is a natural relaxation, where this assumption is not made and the goal is instead to do as well as the best hypothesis in the class. As such, providing a complete characterization for the agnostic case remains an important future direction.

**Beyond worst-case analysis.** Another direction for future research is to analyze structured Stackelberg games in the "beyond worst-case" framework (e.g. Balcan et al. (2018a); Haghtalab et al. (2024); Dick et al. (2020); Balcan et al. (2021)). It has recently been shown that smooth online learning is as easy as distributional learning, with regret bounds scaling with the VC dimension of the hypothesis class (Block et al., 2022). Extending these findings to structured Stackelberg games might lead to improved results both from a statistical and computational point of view.

**Multiple followers and bandit feedback.** Finally, our analysis focuses on the setting where the leader interacts with a single follower, and full feedback is available about the follower's type. Relaxing either of these directions is an important avenue for future research. For the former, techniques that leverage the geometric structure of the problem, as in Personnat et al. (2025) might be useful.

## Acknowledgments

We would like to thank the anonymous reviewers for helpful comments and suggestions. This work was supported in part by the Simons Investigator Award MPS-SICS-00826333, a Microsoft Research Faculty Fellowship, and NSF grant IIS1901403. We are grateful to Brian Hu Zhang for helpful discussions on an earlier version of this paper. For part of this work, KH was a Ph.D. student at Carnegie Mellon University.

## Impact Statement

This paper presents work whose goal is to advance the field of Machine Learning. There are many potential societal consequences of our work, none which we feel must be specifically highlighted here.

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

# A. Applications beyond Stackelberg games

Our results also apply to the problems of learning to bid in simultaneous second-price auctions with side information and Bayesian persuasion with both public and private states. These settings are formally introduced in Balcan et al. (2026); we give an overview here. They are mathematically equivalent to the setting we study in the main body, with the exception of the form that the learner's strategy space takes (detailed below).

## A.1. Simultaneous second-price auctions with side information

This application is an extension of a setting in Daskalakis & Syrgkanis (2022) to incorporate side information. In each round (or historical data point in the distributional setting), bidders simultaneously bid on a set of $m$ items. They receive the bundle of items for which they are the highest bidder, and they pay a price corresponding to the second-highest bid for each item.

The learner in this setting is a bidder whose strategy space consists of a set of possible bids for each item. The bidder's valuation depends on the entire bundle of items they receive, e.g. the bidder may have a low valuation for item A and item B *in silos*, but a high valuation for the combination of the two. In addition to the bundle of items, the bidder's valuation also depends on contextual information, e.g. different seasons or fashion trends may influence a bidder's valuation for a bundle of clothing items.

Other players' bids roughly correspond to different follower types, and we posit a hypothesis class mapping contextual information to different sets of other players' bids. Our results carry over to this application as-is when the space of the other players' bids is discrete; the only difference is that the leader's strategy space is now the bid space $[0,1]^m$ instead of a probability simplex.

## A.2. Bayesian persuasion with both public and private states

In Bayesian persuasion (Kamenica & Gentzkow, 2011), a sender (analogous to the leader in Stackelberg games) strategically reveals payoff-relevant information about an underlying *private* state of the world to a receiver (analogous to the follower), who then takes a payoff-relevant action. This sub-section considers a generalization of Bayesian persuasion where there is some additional payoff-relevant information about the state which is common knowledge to both players. Receiver types map to follower types, and we posit a relationship between the common knowledge and the receiver's type. As is the case in the previous sub-section, our results carry over here as-is, with the sender optimizing over a convex polytope strategy space instead of a probability simplex.

# B. Details and proofs from Section 3

**Theorem 3.5.** *There exists a Stackelberg game $\mathcal{G}$ and a hypothesis class $\mathcal{H}$ such that $\mathrm{Ldim}(\mathcal{H}) = \infty$, but the leader's optimal policy can be determined without learning.*

*Proof.* We construct such a game explicitly. Let the context space be $\mathcal{Z} = \{0,1\} \times [0,1]$ and let the set of follower types be $\{f^{(1)}, f^{(2)}, f^{(3)}, f^{(4)}\}$. Consider the follower types $f^{(1)}$ and $f^{(2)}$ with utility functions $u_{f^{(1)}}, u_{f^{(2)}} : \mathcal{Z} \times \mathcal{A} \times \mathcal{A}_f \to [0,1]$ chosen such that for every context $\mathbf{z} \in \mathcal{Z}$, the leader's optimal action differs, i.e. $\pi_*^{(1)}(\mathbf{z}) \neq \pi_*^{(2)}(\mathbf{z})$, where $\pi_*^{(i)}$, refers to the optimal policy against follower $f^{(i)}$. Concrete payoff matrices for all follower types are shown in Table 1 (the leader is the row player and the follower is the column player).

We now construct the hypothesis class $\mathcal{H} = \{h_{\gamma_1, \gamma_2} : \forall \ 0 < \gamma_1 < 0.5, \ 0.5 \leq \gamma_2 < 1\}$, where

$$
h_{\gamma_1, \gamma_2}(\mathbf{z}) = \begin{cases} f^{(1)} & \text{if } 0 \leq \mathbf{z}[2] < \gamma_1 \\ f^{(2)} & \text{if } \gamma_2 < \mathbf{z}[2] \leq 1 \\ f^{(3)} & \text{if } (\mathbf{z}[1] = 0 \text{ and } \gamma_1 \leq \mathbf{z}[2] < 0.5) \text{ or } (\mathbf{z}[1] = 1 \text{ and } 0.5 \leq \mathbf{z}[2] \leq \gamma_2) \\ f^{(4)} & \text{if } (\mathbf{z}[1] = 1 \text{ and } \gamma_1 \leq \mathbf{z}[2] < 0.5) \text{ or } (\mathbf{z}[1] = 0 \text{ and } 0.5 \leq \mathbf{z}[2] \leq \gamma_2) \end{cases}
$$

Observe that by construction, for any true hypothesis $h_* \in \mathcal{H}$ we have that the optimal policy $\pi_{h^*}$, will satisfy $\pi_{h^*}(\mathbf{z}) = \pi_*^{(1)}(\mathbf{z})$ if $\mathbf{z}[2] < \frac{1}{2}$ and $\pi_{h^*}(\mathbf{z}) = \pi_*^{(2)}(\mathbf{z})$ otherwise. Therefore, no matter which hypothesis is used to predict the follower's type, the utility of the leader will be optimal. On the other hand, determining $h^*$ requires learning the exact cutoff values, $\gamma_1$

|       | $F_1$    | $F_2$    |
|-------|----------|----------|
| $L_1$ | $1, 0.5$ | $0.5, 0$ |
| $L_2$ | $0.25, 1$| $0.75, 0$|

*(a)* Follower type $f^{(1)}$

|       | $F_1$    | $F_2$      |
|-------|----------|------------|
| $L_1$ | $1, 0$   | $0.5, 0.5$ |
| $L_2$ | $0.25, 0$| $0.75, 1$  |

*(b)* Follower type $f^{(2)}$

|       | $F_1$                 | $F_2$            |
|-------|-----------------------|------------------|
| $L_1$ | $1, 1 - \mathbf{z}[1]$| $0.5, 0.5$       |
| $L_2$ | $0.25, 0.5$           | $0.75, \mathbf{z}[1]$ |

*(c)* Follower type $f^{(3)}$

|       | $F_1$             | $F_2$                     |
|-------|-------------------|---------------------------|
| $L_1$ | $1, \mathbf{z}[1]$| $0.5, 0.5$                |
| $L_2$ | $0.25, 0.5$       | $0.75, 1 - \mathbf{z}[1]$ |

*(d)* Follower type $f^{(4)}$

*Table 1.* Utility tables for Theorem 3.5

and $\gamma_2$. This corresponds to learning the class of linear thresholds, which has infinite Littlestone dimension (cf. Example 21.4 in (Shalev-Shwartz & Ben-David, 2014)). Therefore, $\mathrm{Ldim}(\mathcal{H}) = \infty$. $\qquad\square$

**Theorem 3.8.** *For any hypothesis class $\mathcal{H}$ and Stackelberg game $\mathcal{G}$, there exists an adversarial sequence of contexts and follower types such that any deterministic algorithm suffers at least $\mathrm{SLdim}_{\mathcal{G}}(\mathcal{H}) - \epsilon$ regret, for any $\epsilon > 0$.*

*Proof.* We prove by induction on the depth $d$ of shattered trees that across $d$ timesteps the learner can be forced to incur regret at least as much as the weight of the tree's root node. Since the SL dimension is defined as the supremum root node weight over all finite-depth trees, proving this suffices to show the desired claim. In case the supremum in the definition of $\mathrm{SLdim}_{\mathcal{G}}(\mathcal{H})$ is not achieved by any finite tree, there will always be an additive error of $\epsilon$ that will get arbitrarily small as $d \to \infty$.

(Base Case) We consider a shattered tree of depth $1$. Then

$$
\rho_\varnothing = \inf_{\mathbf{x} \in \Delta(\mathcal{A})} \max_{i \in \mathcal{H}(\mathbf{z}_\varnothing)} r(\mathbf{z}_\varnothing, \mathbf{x}, f^{(i)})
$$

$$
= \inf_{\mathbf{x} \in \Delta(\mathcal{A})} \max_{i \in \mathcal{H}(\mathbf{z}_\varnothing)} \left[ u(\mathbf{z}_\varnothing, \pi_{h^*}(\mathbf{z}_\varnothing), b_{f^{(i)}}(\mathbf{z}_\varnothing, \pi_{h^*}(\mathbf{z}_\varnothing))) - u(\mathbf{z}_\varnothing, \mathbf{x}, b_{f^{(i)}}(\mathbf{z}_\varnothing, \mathbf{x})) \right].
$$

So $\rho_\varnothing$ corresponds to the min-max value of the one-shot Stackelberg game and therefore, to a lower bound on the instantaneous regret of the learner.

(Inductive step) We assume that for any tree of depth $d$ that is shattered by $\mathcal{H}$ under $\mathcal{G}$, the weight of the root node lower bounds the learner's cumulative regret across $d$ timesteps. We now consider a tree $\mathcal{T}_{d+1}$ of depth $d + 1$ that is shattered by $\mathcal{H}$ under Stackelberg game $\mathcal{G}$. We will prove that for such a tree, the adversary can guarantee regret at least $\rho_\varnothing$, where $\mathbf{z}_\varnothing$ is the root node of this tree. By Definition 3.6, we know that $\rho_\varnothing = \inf_{\mathbf{x} \in \Delta(\mathcal{A})} \max_{i \in \mathcal{H}(\mathbf{z}_\varnothing)} \left[ r(\mathbf{z}_\varnothing, \mathbf{x}, f^{(i)}) + \rho_{\varnothing i} \right]$. Let $\mathbf{x}_* \in \Delta(\mathcal{A})$ and $i_* \in [K]$ be the respective arginf and argmax values in the above expression.

If the learner chooses to play $\mathbf{x}_*$ at this round, then by definition the adversary can guarantee instantaneous regret equal to $r(\mathbf{z}_\varnothing, \mathbf{x}_*, f^{(i_*)})$ by selecting the true follower type to be $i_*$ (which is realizable by $\mathcal{H}$ by definition of a shattered tree). If this happens, then by the inductive hypothesis the adversary can guarantee regret $\rho_{i_*}$ in the next $d$ timesteps and regret $\rho_\varnothing$ overall.

We will now prove the same even when the learner plays a strategy different from $\mathbf{x}_*$. Say that there exists $\tilde{\mathbf{x}} \in \Delta(\mathcal{A})$ such that the learner can guarantee instantaneous regret $r(\mathbf{z}_\varnothing, \tilde{\mathbf{x}}, f^{(\tilde{i})}) < r(\mathbf{z}_\varnothing, \mathbf{x}_*, f^{(i_*)})$, where $\tilde{i} = \mathrm{argmax}_{i \in \mathcal{H}(\mathbf{z}_\varnothing)} \left[ r(\mathbf{z}_\varnothing, \tilde{\mathbf{x}}, f^{(i)}) + \rho_i \right]$. There are two cases.

(Case 1) Suppose $\tilde{i} = i_*$. Then we know that

$$
\inf_{\mathbf{x} \in \Delta(\mathcal{A})} \max_{i \in \mathcal{H}(\mathbf{z}_\varnothing)} \left[ r(\mathbf{z}_\varnothing, \mathbf{x}, f^{(i)}) + \rho_i \right] = \inf_{\mathbf{x} \in \Delta(\mathcal{A})} \left[ r(\mathbf{z}_\varnothing, \mathbf{x}, f^{(i_*)}) + \rho_{i_*} \right].
$$

Since $\mathbf{x}_*$ achieves the infimum value

$$r(\mathbf{z}_\varnothing, \mathbf{x}_*, f^{(i_*)}) + \rho_{i_*} \leq r(\mathbf{z}_\varnothing, \tilde{\mathbf{x}}, f^{(i_*)}) + \rho_{i_*} \quad \forall \tilde{\mathbf{x}} \in \Delta(\mathcal{A})$$
$$\implies r(\mathbf{z}_\varnothing, \mathbf{x}_*, f^{(i_*)}) \leq r(\mathbf{z}_\varnothing, \tilde{\mathbf{x}}, f^{(i_*)}) \quad \forall \tilde{\mathbf{x}} \in \Delta(\mathcal{A}),$$

which leads to a contradiction.

(Case 2) Suppose $\tilde{i} \neq i_*$. Then we have that

$$\inf_{\mathbf{x} \in \Delta(\mathcal{A})} \max_{i \in \mathcal{H}(\mathbf{z}_\varnothing)} \left[ r(\mathbf{z}_\varnothing, \mathbf{x}, f^{(i)}) + \rho_i \right] \leq \max_{i \in \mathcal{H}(\mathbf{z}_\varnothing)} \left[ r(\mathbf{z}_\varnothing, \mathbf{x}', f^{(i)}) + \rho_i \right] \quad \forall \mathbf{x}' \in \Delta(\mathcal{A})$$

Taking $\mathbf{x}' = \tilde{\mathbf{x}}$, we get

$$\inf_{\mathbf{x} \in \Delta(\mathcal{A})} \max_{i \in \mathcal{H}(\mathbf{z}_\varnothing)} \left[ r(\mathbf{z}_\varnothing, \mathbf{x}, f^{(i)}) + \rho_i \right] \leq \max_{i \in \mathcal{H}(\mathbf{z}_\varnothing)} \left[ r(\mathbf{z}_\varnothing, \tilde{\mathbf{x}}, f^{(i)}) + \rho_i \right]$$
$$\implies r(\mathbf{z}_\varnothing, \mathbf{x}_*, f^{(i_*)}) + \rho_{i_*} \leq r(\mathbf{z}_\varnothing, \tilde{\mathbf{x}}, f^{(\tilde{i})}) + \rho_{\tilde{i}}$$
$$\implies r(\mathbf{z}_\varnothing, \mathbf{x}_*, f^{(i_*)}) - r(\mathbf{z}_\varnothing, \tilde{\mathbf{x}}, f^{(\tilde{i})}) \leq \rho_{\tilde{i}} - \rho_{i_*}$$

We denote $\ell := r(\mathbf{z}_\varnothing, \mathbf{x}_*, f^{(i_*)}) - r(\mathbf{z}_\varnothing, \tilde{\mathbf{x}}, f^{(\tilde{i})})$ and by definition of $\tilde{\mathbf{x}}$ we know that $\ell > 0$. We get that $\rho_{\tilde{i}} \geq \rho_{i_*} + \ell$. Now by the boundedness of the utility function, we know that if $\rho_{i_*} + \ell \geq d$, then the above inequality leads to a contradiction (the learner cannot incur strictly more than $d$ regret over $d$ timesteps).

Otherwise, we have a subtree with root node weight at least $\rho_{i_*} + \ell$ that is shattered by the induced hypothesis space $\mathcal{H}^{(\mathbf{z}_\varnothing \to f^{(\tilde{i})})}$. By the inductive hypothesis we know that the adversary can force at least $\rho_{i_*} + \ell$ regret over the remaining $d$ timesteps and therefore at least $r(\mathbf{z}_\varnothing, \tilde{\mathbf{x}}, f^{(\tilde{i})}) + \rho_{i_*} + \ell = r(\mathbf{z}_\varnothing, \mathbf{x}_*, f^{(i_*)}) + \rho_{i_*} = \rho_\varnothing$ regret over $d+1$ timesteps, as desired. $\qquad \square$

**Theorem 3.9.** *For any Stackelberg game $\mathcal{G}$ and every hypothesis class $\mathcal{H}$, the regret of Algorithm 1 is at most $\mathrm{SLdim}_\mathcal{G}(\mathcal{H})$.*

*Proof.* We will show that at each timestep $t$, if the learner incurred instantaneous regret $r_t = r(\mathbf{z}_t, \mathbf{x}_t, f_t)$, then the the SL dimension must have decreased by at least $r_t$. In other words, $\mathrm{SLdim}_\mathcal{G}(V_{t+1}) \leq \mathrm{SLdim}_\mathcal{G}(V_t) - r_t$, where $V_t$ is the set of hypotheses consistent with the history up to timestep $t$. We have that the regret at timestep $t$ is

$$r_t = u(\mathbf{z}_t, \mathbf{x}_*^{(i, \mathbf{z}_t)}, b_{f_t}(\mathbf{z}_t, \mathbf{x}_*^{(i, \mathbf{z}_t)})) - u(\mathbf{z}_t, \mathbf{x}_t, b_{f_t}(\mathbf{z}_t, \mathbf{x}_t)) + (\mathrm{SLdim}_\mathcal{G}(V_t^{(\mathbf{z}_t \to f_t)}) - \mathrm{SLdim}_\mathcal{G}(V_t^{(\mathbf{z}_t \to f_t)}))$$
$$\leq \max_{i \in V_t(\mathbf{z}_t)} \left[ u(\mathbf{z}_t, \mathbf{x}_*^{(i, \mathbf{z}_t)}, b_{f_t}(\mathbf{z}_t, \mathbf{x}_*^{(i, \mathbf{z}_t)})) - u(\mathbf{z}_t, \mathbf{x}_t, b_{f_t}(\mathbf{z}_t, \mathbf{x}_t)) + \mathrm{SLdim}_\mathcal{G}(V_t^{(\mathbf{z}_t \to i)}) \right] - \mathrm{SLdim}_\mathcal{G}(V_t^{(\mathbf{z}_t \to f_t)})$$
$$= \inf_{\mathbf{x} \in \Delta(\mathcal{A})} \max_{i \in V_t(\mathbf{z}_t)} \left[ u(\mathbf{z}_t, \mathbf{x}_*^{(i, \mathbf{z}_t)}, b_{f_t}(\mathbf{z}_t, \mathbf{x}_*^{(i, \mathbf{z}_t)})) - u(\mathbf{z}_t, \mathbf{x}, b_{f_t}(\mathbf{z}_t, \mathbf{x})) + \mathrm{SLdim}_\mathcal{G}(V_t^{(\mathbf{z}_t \to i)}) \right] - \mathrm{SLdim}_\mathcal{G}(V_{t+1})$$
$$\text{(by definition of Algorithm 1)}$$
$$\leq \sup_{\mathbf{z} \in \mathcal{Z}} \inf_{\mathbf{x} \in \Delta(\mathcal{A})} \max_{i \in V_t(\mathbf{z})} \left[ u(\mathbf{z}, \mathbf{x}_*^{(i, \mathbf{z})}, b_{f_t}(\mathbf{z}, \mathbf{x}_*^{(i, \mathbf{z})})) - u(\mathbf{z}, \mathbf{x}, b_{f_t}(\mathbf{z}, \mathbf{x})) + \mathrm{SLdim}_\mathcal{G}(V_t^{(\mathbf{z} \to i)}) \right] - \mathrm{SLdim}_\mathcal{G}(V_{t+1})$$
$$= \mathrm{SLdim}_\mathcal{G}(V_t) - \mathrm{SLdim}_\mathcal{G}(V_{t+1}),$$

where $\mathbf{x}_*^{(i, \mathbf{z})}$ is the leader's optimal strategy against follower $f^{(i)}$ for context $\mathbf{z}$. $\qquad \square$

**Lemma 3.10.** *For any hypothesis class $\mathcal{H}$ and any Stackelberg game $\mathcal{G}$, $\mathrm{SLdim}_\mathcal{G}(\mathcal{H}) \leq \mathrm{Ldim}(\mathcal{H})$.*

*Proof.* We assume for the sake of contradiction that there exists a hypothesis class with $\mathrm{Ldim}(\mathcal{H}) = d$ and a Stackelberg game $\mathcal{G}$, such that $\mathrm{SLdim}_\mathcal{G}(\mathcal{H}) > d$. According to Definition B.5, this implies that there exists a shattered tree, whose minimum-weighted root-to-leaf path has weight strictly greater than $d$. This implies that all paths in the tree have cumulative weight strictly greater than $d$. Given that the weight of each edge in the tree is at most $1$, the length of the minimum path (and therefore of all other paths in the tree) must be of length at least $d+1$. This implies that there exists a tree of depth $d+1$ that is shattered by $\mathcal{H}$, which leads to a contradiction. $\qquad \square$

*Table 2.* Utility tables for the leader (row player) and each follower type (column player)

*(a)* Follower type $f^{(1)}$

|       | $F_1$    | $F_2$ |
|-------|----------|-------|
| $L_1$ | $1, 1/2$ | $0, 0$ |
| $L_2$ | $0, 0$   | $1, 0$ |

*(b)* Follower type $f^{(2)}$

|       | $F_1$                        | $F_2$                        |
|-------|------------------------------|------------------------------|
| $L_1$ | $1, \mathbb{1}_{\{z \neq 4\}}/2$ | $0, 0$                   |
| $L_2$ | $0, 0$                       | $1, \mathbb{1}_{\{z = 4\}}/2$ |

*(c)* Follower type $f^{(3)}$

|       | $F_1$                        | $F_2$                        |
|-------|------------------------------|------------------------------|
| $L_1$ | $1, \mathbb{1}_{\{z = 3\}}/2$ | $0, 0$                   |
| $L_2$ | $0, 0$                       | $1, \mathbb{1}_{\{z \neq 3\}}/2$ |

**Theorem 3.11.** *There exists a hypothesis class $\mathcal{H}$ and Stackelberg game $\mathcal{G}$ such that running SOA on $\mathcal{H}$ results in suboptimal utility for the leader.*

*Proof.* Consider $\mathcal{Z} = \{1, 2, 3, 4\}$ and followers $\{f^{(1)}, f^{(2)}, f^{(3)}\}$. We define $\mathcal{H} = \{h_\sigma(z) := \mathbb{1}\{z \neq 4\} \cdot f^{(\sigma(z))} + \mathbb{1}\{z = 4\} \cdot f^{(3)}\}_{\sigma \in \Sigma} \cup \{h, h'\}$, where $\Sigma$ is the set of all permutations of the set $\{1, 2, 3\}$ and $h, h'$ are defined as follows.

$$
h(z) = \begin{cases}
f^{(3)} & \text{if } z = 1 \\
f^{(2)} & \text{if } z = 2 \\
f^{(1)} & \text{if } z = 3 \\
f^{(2)} & \text{if } z = 4
\end{cases}
$$

$$
h'(z) = \begin{cases}
f^{(3)} & \text{if } z = 1 \\
f^{(1)} & \text{if } z = 2 \\
f^{(2)} & \text{if } z = 3 \\
f^{(2)} & \text{if } z = 4
\end{cases}
$$

Notice that the Littlestone dimension of this hypothesis class is at least 2. This follows from the $\mathcal{H}$-shattered tree construction shown in Figure 4.

Consider now a Stackelberg game with utility functions $u_{f^{(1)}}, u_{f^{(2)}}, u_{f^{(3)}}$, defined such that if $z = 1$ or $z = 2$ then $\pi_*^{(1)}(z) = \pi_*^{(2)}(z) \neq \pi_*^{(3)}(z)$, if $z = 3$ then $\pi_*^{(1)}(3) = \pi_*^{(2)}(3) = \pi_*^{(3)}(3)$, and if $z = 4$, then $\pi_*^{(1)}(4) \neq \pi_*^{(2)}(4) = \pi_*^{(3)}(4)$. Here, we let $\pi_*^{(i)} : \mathcal{Z} \to \Delta(\mathcal{A})$ denote the optimal leader's policy against follower type $f^{(i)}$. Concretely, consider a Stackelberg game with the leader being the row player and each follower being the column player, as shown in Table 2.

We now show that the $\text{SLdim}_\mathcal{G}(\mathcal{H}) = 3/4$. We will consider any possible SL tree. First, note that without loss of generality we can consider trees that do not include the context $z = 4$ since for $z = 4$, $\pi_*^{(2)}(4) = \pi_*^{(3)}(4)$, where $f^{(2)}$ and $f^{(3)}$ are the only possible follower types predicted by hypotheses in $\mathcal{H}$ for this context. Similarly, for $z = 3$, since the leader's optimal strategy against all follower types is identical ($\pi_*^{(1)}(3) = \pi_*^{(2)}(3) = \pi_*^{(3)}(3)$).

So it suffices to consider contexts 1 and 2. In Figure 5 we show the two shattered SL tree constructions with $z = 1$ and $z = 2$ as the root nodes respectively. We start with the tree in Figure 5a. For $z = 2$, its not hard to show that $\inf_\mathbf{x} \max_{j \in \{1, 3\}} r(\mathbf{x}, 2, f^{(j)}) = \inf_\mathbf{x} \max_{j \in \{2, 3\}} r(\mathbf{x}, 2, f^{(j)}) = 1/2$, achieved at $\mathbf{x} = (1/2, 1/2)$. For the root node $z = 1$, we have that $\inf_\mathbf{x} \max\{r(\mathbf{x}, 1, f^{(1)}) + 1/2, \ r(\mathbf{x}, 1, f^{(2)}) + 1/2, \ r(\mathbf{x}, 1, f^{(3)})\} = 3/4$ achieved at $\mathbf{x} = (3/4, 1/4)$. Therefore, the weight of the root node is $3/4$. A similar analysis shows that the root node weight of the SL-tree in Figure 5b is $3/4$. Therefore, $\text{SLdim}_\mathcal{G}(\mathcal{H})$ (and also the worst-case cumulative regret) equals $3/4$.

We consider the case, where the adversary present context 1 followed by context 2. We go through the SOA outputs per timestep below.

- $(t = 1)$ This node has three children corresponding to follower types $f^{(1)}, f^{(2)}, f^{(3)}$, with $f^{(1)}$ and $f^{(2)}$ leading to a full binary subtree of maximal depth 1 each. To see this, notice that (1) neither $h$ nor $h'$ are consistent with the mapping of context $z = 1$ to follower type $f^{(1)}$ or $f^{(2)}$, and (2) $\mathcal{H} \setminus \{h, h'\}$ consists of all permutations mapping contexts $\{1, 2, 3\}$ to follower types $\{f^{(1)}, f^{(2)}, f^{(3)}\}$. Therefore, at most two of these contexts can exist in any root-to-leaf path of any $\mathcal{H}$-shattered tree. The child corresponding to follower type $f^{(3)}$ leads to a full binary subtree of maximal depth at least 2, as shown in Figure 4. So, the predicted follower type is $f_1 = f^{(3)}$ and the strategy played is $x_1 = \pi_*^{(3)}(1)$. However, the true follower type is $f_1^* = f^{(1)}$ (instantaneous regret equals 1).

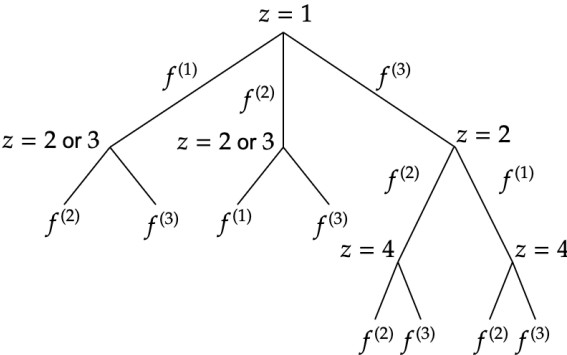

*Figure 4.* Example construction of an $\mathcal{H}$-shattered Littlestone tree. Each node represents a context in $\mathcal{Z} = \{1, 2, 3, 4\}$ and each edge corresponds to a follower type.

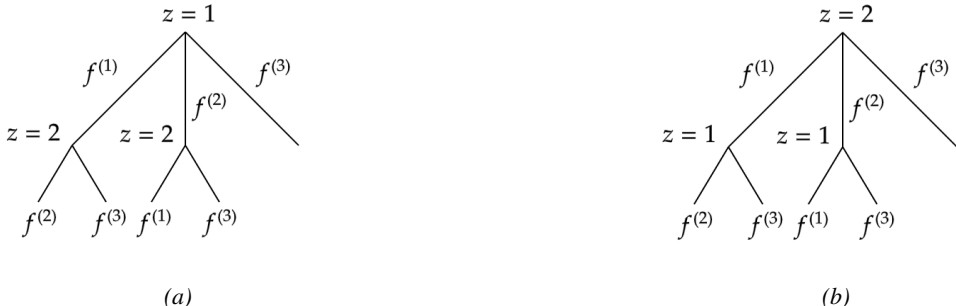

*Figure 5.* $\mathcal{H}$-shattered SL-Tree with (a) $z = 1$ as the root node and (b) $z = 2$ as the root node.

- ($t = 2$) This node has two children mapping $z = 2$ to follower types $f^{(2)}$ and $f^{(3)}$. No subsequent context can lead to a full-binary tree that is consistent with any of these assignments, so SOA tie-breaks. The adversary in this case can always force a mistake by claiming the correct follower type to be the one not predicted by the algorithm. So if SOA predicts $f_2 = f^{(2)}$, and plays strategy $x_2 = \pi_*^{(2)}(2)$ the adversary can choose the true follower type to be $f_2^* = f^{(3)}$ (instantaneous regret equals 1).

This leads to SOA mispredicting the follower type and getting suboptimal utility in both rounds. Since the Stackelberg Littlestone dimension is $3/4$ and we have shown that for the above sequence of contexts running SOA yields regret 2, this concludes our proof. □

## B.1. A technical subtlety

An important component of our analysis is that the leader must be able to compute the optimal strategy for a given context and follower type. For this, we state that for any context $\mathbf{z} \in \mathcal{Z}$ the leader incurs negligible regret by restricting herself to a subset $\mathcal{E}_{\mathbf{z}} \subseteq \Delta(\mathcal{A})$ which is finite and computable. To formally state our result we present the notion of a *contextual best-response region*, which was first introduced in (Harris et al., 2024) and which we repeat here for completeness.

**Definition B.1** (Contextual Follower Best-Response Region). For context $\mathbf{z}$, follower type $f^{(i)}$, and follower action $a_f \in \mathcal{A}_f$, let $X_{\mathbf{z}}(f^{(i)}, a_f) \subseteq \mathcal{X}$ denote the leader's mixed strategies such that a follower of type $f^{(i)}$ best-responds to all $x \in X_{\mathbf{z}}(f^{(i)}, a_f)$ by playing action $a_f$, that is, $X_{\mathbf{z}}(f^{(i)}, a_f) = \{\mathbf{x} \in \mathcal{X} : b_{f^{(i)}}(\mathbf{z}, \mathbf{x}) = a_f\}$

Notice that any possible follower best-response region is (1) a subset of $\Delta(\mathcal{A})$ and (2) the intersection of finitely-many halfspaces. Therefore, every $X_{\mathbf{z}}(f^{(i)}, a_f)$ is convex and bounded. However, the leader may not be able to optimize over the extreme points of a best-response region, as it may not be closed due to the followers' tie-breaking.

**Definition B.2** ($\delta$-approximate extreme points)**.** Fix a context $\mathbf{z} \in \mathcal{Z}$ and consider the set of all non-empty contextual best-response regions. For $\delta > 0$, $\mathcal{E}_{\mathbf{z}}(\delta)$ is the set of leader mixed strategies such that for all best-response functions $\sigma$ and any $\mathbf{x} \in \Delta(\mathcal{A})$ that is an extreme point of $\text{cl}(\mathcal{X}_{\mathbf{z}}(\sigma))$, $\mathbf{x} \in \mathcal{E}_{\mathbf{z}}(\delta)$ if $\mathbf{x} \in \mathcal{X}_{\mathbf{z}}(\sigma)$. Otherwise there is some $\mathbf{x}' \in \mathcal{E}_{\mathbf{z}}(\delta)$ such that $\mathbf{x}' \in \mathcal{X}_{\mathbf{z}}(\sigma)$ and $\|\mathbf{x}' - \mathbf{x}\|_1 \leq \delta$.

Notice that for sufficiently small $\delta$ i.e. $\delta = \mathcal{O}(\frac{1}{T})$, we get that the additional regret of the leader by choosing actions over $\mathcal{E}^{(i)}$ rather than $X(f^{(i)}, a_f)$ is negligible. Therefore, we use the shorthand $\mathcal{E}^{(i)} := \mathcal{E}^{(i)}(\delta)$ for the rest of the section. The following lemma from (Harris et al., 2024) formalizes this notion.

**Lemma B.3.** *For any sequence of followers $f_1, \ldots f_T$ and any leader policy $\pi$, there exists a policy $\pi^{(\mathcal{E})} : \mathcal{Z} \to \cup_{\mathbf{z} \in \mathcal{Z}} \mathcal{E}_{\mathbf{z}}$ that, when given context $\mathbf{z}$, plays a mixed strategy in $\mathcal{E}_{\mathbf{z}}$ and guarantees that $\sum_{t=1}^{T} u(\mathbf{z}_t, \pi(\mathbf{z}_t), b_{f_t}(\mathbf{z}_t, \pi(\mathbf{z}_t))) - u(\mathbf{z}_t, \pi^{(\mathcal{E})}(\mathbf{z}_t), b_{f_t}(\mathbf{z}_t, \pi^{(\mathcal{E})}(\mathbf{z}_t))) \leq 1$. Moreover, the same result holds in expectation over any distribution over follower types.*

Given Lemma B.3 and for ease of notation, in this paper we assume that the leader will always set the parameter $\delta$, such that it does not affect our regret bounds.

## B.2. Alternate definition of the Stackelberg Littlestone dimension

In this section we provide an alternate definition of the Stackelberg Littlestone dimension that can be defined for infinite trees. In Theorem B.6 we prove that for finite trees, this is equivalent to Definition 3.7.

**Definition B.4** ((Edge-weighted) SL tree)**.** An edge-weighted SL tree, $\mathcal{T}$, is a rooted tree, in which each internal node is labeled with an instance in $\mathcal{Z}$ and each edge corresponds to a label in $[K]$. For each node, $\mathbf{z}_s \in \mathcal{T}$ with children $\{\mathbf{z}_{si}\}_{i \in M \subseteq [K]}$ the outgoing edges have weights, $\{\nu_s^{(i)}\}_{i \in M}$, such that for every $\mathbf{x} \in \Delta(\mathcal{A})$ there exists $i \in M$ such that $r(\mathbf{z}_s, \mathbf{x}, f^{(i)}) \geq \nu_s^{(i)}$. We say that the tree is *shattered* by a hypothesis class $\mathcal{H}$ under Stackelberg game $\mathcal{G}$ if for every path that traverses nodes $\mathbf{z}_{\varnothing}, \ldots \mathbf{z}_{s_{\leq d}}$, there exists $h \in \mathcal{H}$ such that the label of the edge in the path leaving $\mathbf{z}_{s_{\leq i}}$ is $h(\mathbf{z}_{s_{\leq i}})$ for every $i$.

**Definition B.5** ((Edge-weighted) SL dimension)**.** The SL dimension of a hypothesis class $\mathcal{H}$ with respect to a Stackelberg game, $\mathcal{G}$, is defined as $\text{SLdim}_{\mathcal{G}}(\mathcal{H}) = \sup\{\kappa \in \mathbb{R}_{\geq 0} :$ exists finite edge-weighted SL tree shattered by $\mathcal{H}$ under $\mathcal{G}$ where all root-to-leaf paths have cumulative weight at least $\kappa\}$.

**Theorem B.6.** *For finite trees, Definition 3.7 and Definition B.5 coincide.*

*Proof.* It suffices to show that for any finite depth $d$ edge-weighted tree with all root-to-leaf paths having cumulative weight at least $\kappa$ can be shattered under Definition B.4 if and only if there exists a tree with root node weight $\kappa$ that can be shattered under Definition 3.6. We proceed by induction on the depth.

(Base Case) Consider a depth-1 tree, $\mathcal{T}_1$, with root node $\mathbf{z}_{\varnothing}$. In the node-weight formulation, $\rho_{\varnothing} = \inf_{\mathbf{x} \in \Delta(\mathcal{A})} \max_{i \in \mathcal{H}(\mathbf{z}_{\varnothing})} r(\mathbf{z}_{\varnothing}, \mathbf{x}, f^{(i)})$. In the edge-weight formulation, the valid edge-weight assignments are $\mathcal{N}_{\mathbf{z}_{\varnothing}} = \left\{N = \{\nu_{\varnothing}^{(i)}\}_{i \in \mathcal{H}(\mathbf{z}_{\varnothing})} : \forall \mathbf{x} \exists i \ r(\mathbf{z}_{\varnothing}, \mathbf{x}, f^{(i)}) \geq \nu_{\varnothing}^{(i)}\right\}$. Their minimal path weight is $\max_{N \in \mathcal{N}_{\mathbf{z}_{\varnothing}}} \min_{i \in \mathcal{H}(\mathbf{z}_{\varnothing})} \nu_{\varnothing}^{(i)}$.

We will first show that $\inf_{\mathbf{x} \in \Delta(\mathcal{A})} \max_{i \in \mathcal{H}(\mathbf{z}_{\varnothing})} r(\mathbf{z}_{\varnothing}, \mathbf{x}, f^{(i)}) \geq \max_{N \in \mathcal{N}_{\mathbf{z}_{\varnothing}}} \min_{i \in \mathcal{H}(\mathbf{z}_{\varnothing})} \{\nu_{\varnothing}^{(i)} : \nu_{\varnothing}^{(i)} \in N\}$. Fix some $N \in \mathcal{N}_{\mathbf{z}_{\varnothing}}$. Then for every $\mathbf{x} \in \Delta(\mathcal{A})$, $\exists i \in \mathcal{H}(\mathbf{z}_{\varnothing})$ such that $r(\mathbf{z}_{\varnothing}, \mathbf{x}, f^{(i)}) \geq \nu_{\varnothing}^{(i)}$. This means that

$$\max_{i \in \mathcal{H}(\mathbf{z}_{\varnothing})} r(\mathbf{z}_{\varnothing}, \mathbf{x}, f^{(i)}) \geq \min_{i \in \mathcal{H}(\mathbf{z}_{\varnothing})} \{\nu_{\varnothing}^{(i)} : \nu_{\varnothing}^{(i)} \in N\} \quad \forall \mathbf{x} \in \Delta(\mathcal{A}) \ \forall N \in \mathcal{N}_{\mathbf{z}_{\varnothing}}.$$

Taking the infimum over $\mathbf{x}$ and the maximum over $N \in \mathcal{N}_{\mathbf{z}_{\varnothing}}$ yields the inequality.

For the reverse inequality, let $\mathbf{x}^* \in \arg\inf_{\mathbf{x} \in \Delta(\mathcal{A})} \max_{i \in \mathcal{H}(\mathbf{z}_{\varnothing})} r(\mathbf{z}_{\varnothing}, \mathbf{x}, f^{(i)})$ and $N^* = \{\nu_{\varnothing}^{(i)} := \arg\max_{i \in \mathcal{H}(\mathbf{z}_{\varnothing})} r(\mathbf{z}_{\varnothing}, \mathbf{x}^*, f^{(i)})\}_{i \in [K]}$. We argue that $N^* \in \mathcal{N}_{\mathbf{z}_{\varnothing}}$, i.e., that

$$\forall \mathbf{x} \in \Delta(\mathcal{A}) \ \exists i \in \mathcal{H}(\mathbf{z}_{\varnothing}) \ \text{such that} \ r(\mathbf{z}_{\varnothing}, \mathbf{x}, f^{(i)}) \geq \arg\max_{i \in \mathcal{H}(\mathbf{z}_{\varnothing})} r(\mathbf{z}, \mathbf{x}^*, f^{(i)})$$

which holds since $\max_{i \in \mathcal{H}(\mathbf{z}_{\varnothing})} r(\mathbf{z}_{\varnothing}, \mathbf{x}, f^{(i)}) \geq \inf_{\mathbf{x} \in \Delta(\mathcal{A})} \max_{i \in \mathcal{H}(\mathbf{z}_{\varnothing})} r(\mathbf{z}_{\varnothing}, \mathbf{x}, f^{(i)})$.

(Inductive Step) Assume that the equivalence holds for all trees of depth $d-1$. Consider such a depth-$d$ tree, $\mathcal{T}_d$, with root node weight $\mathbf{z}_\varnothing$ and children $\mathbf{z}_{\varnothing i}$ for $i \in \mathcal{H}(\mathbf{z}_\varnothing)$. We want to show that the root node weight $\rho_\varnothing$ defined as in Definition 3.6 is equal to the weight of the minimum path in the tree, where the edge weights are defined as in Definition B.4. Concretely, we want to show that

$$\inf_{\mathbf{x}\in\Delta(\mathcal{A})} \max_{j\in\mathcal{H}(\mathbf{z}_\varnothing)} \left[ r(\mathbf{z}_\varnothing, \mathbf{x}, f^{(j)}) + \rho_{\varnothing j} \right] = \sup_{\mathcal{N}_d} \inf_{s\in S_d} \left[ \sum_{i=0}^{d} \nu_{s[:i]}^{(s[i+1])} \right],$$

where $\mathcal{N}_d$ is the collection of sets of all possible edge weights, one for each node in the tree of depth $d$. This holds if and only if

$$\inf_{\mathbf{x}\in\Delta(\mathcal{A})} \max_{j\in\mathcal{H}(\mathbf{z}_\varnothing)} \left[ r(\mathbf{z}_\varnothing, \mathbf{x}, f^{(j)}) + \rho_{\varnothing j} \right] = \max_{\mathcal{N}_0} \min_{i\in\mathcal{H}(\mathbf{z}_\varnothing)} \left[ \nu_\varnothing^{(s[1])} + \max_{\mathcal{N}_{d-1}} \inf_{s\in S_{d-1}} \sum_{i=1}^{d} \nu_{s[:i]}^{(s[i+1])} \right]$$

We know that for any $j \in \mathcal{H}(\mathbf{z}_\varnothing)$ and $s \in S_d$ such that $s[1] = j$, $\rho_{\varnothing j} = \max_{\mathcal{N}_{d-1}} \inf_{s\in S_{d-1}} \sum_{i=1}^{d} \nu_{s[:i]}^{(s[i+1])}$ by the inductive hypothesis. Therefore, using an argument similar to the base case, the equation above holds. $\square$

# C. Details and proofs from Section 4

In our work, we consider two variants of sample complexity, PAC cut-off sample complexity (Definition 4.1) and PAC sample complexity defined as follows.

**Definition C.1** (PAC sample complexity). The PAC sample complexity of a function class $\mathcal{H}$ w.r.t. a Stackelberg game $\mathcal{G}$ is defined as $m(\mathcal{H}, \mathcal{G}; \epsilon, \delta) = \inf_A m_A(\mathcal{H}, \mathcal{G}; \epsilon, \delta)$, where the infimum is over all learning algorithms. $m_A(\mathcal{H}, \mathcal{G}; \epsilon, \delta)$ is the smallest integer such that for any $m \geq m_A(\mathcal{H}, \mathcal{G}; \epsilon, \delta)$, every distribution $\mathcal{D}$ on $\mathcal{Z}$, and any true hypothesis $h^* \in \mathcal{H}$, the expected loss of $A$, $\mathbb{E}_{\mathbf{z}\sim\mathcal{D}}\left[ r(\mathbf{z}, A(\mathbf{z}; S), f^{(h^*(\mathbf{z}))}) \right]$, is at most $\epsilon$ with probability at least $1 - \delta$, where $S = \{(\mathbf{z}_i, f^{(h^*(\mathbf{z}_i))})\}_{i\in[m]}$ and $\mathbf{z}_i \sim \mathcal{D}$.

According to the following lemma, any bound with respect to the PAC cut-off complexity immediately implies a bound on the PAC sample complexity as well.

**Lemma C.2** (Lemma 1 in (Attias et al., 2023)). *For every $\epsilon, \delta \in (0,1)^2$ and every $\mathcal{H} \subseteq [0,1]^{\mathcal{Z}}$, it holds that*

$$m(\mathcal{H}; \sqrt{\epsilon}, \delta, \sqrt{\epsilon}) \leq m(\mathcal{H}; \epsilon, \delta) \leq m(\mathcal{H}, \epsilon/2, \delta, \epsilon/2).$$

**Theorem 4.4.** *Let $A$ be any learning algorithm and $\epsilon, \delta, \gamma \in (0,1)$ such that $\delta < \epsilon$, then*

$$m_A(\mathcal{H}, \mathcal{G}; \epsilon, \delta, \gamma) \geq \Omega\left( \frac{\mathrm{SNdim}_\mathcal{G}^{(\gamma)}(\mathcal{H}) + \log(1/\delta)}{\epsilon} \right).$$

*Proof.* Let $n_0 = \mathrm{SNdim}_\mathcal{G}^{(\gamma)}(\mathcal{H})$. Then we know that for any $1 \leq n \leq n_0$ there exists a set $S_n \in \mathcal{Z}^n$ and functions $g_0, g_1 : \mathcal{Z} \to \Delta(\mathcal{A})$ such that for any set $T \subseteq S$ there exists an $h$ that agrees with $g_0$ on all $\mathbf{z} \in T$ and with $g_1$ for any $\mathbf{z} \in S \setminus T$. We will show that for a given learning algorithm $A : (\mathcal{Z} \times [K])^* \to \Delta(\mathcal{A})^{\mathcal{Z}}$, there exists a distribution $\mathcal{D}$ on $\mathcal{Z}$ such that the algorithm requires at least $C_0 \frac{\mathrm{SNdim}_\mathcal{G}^{(\gamma)}(\mathcal{H})+\log(\frac{1}{\delta})}{\epsilon}$ samples. To do that, we will construct a distribution with support the $n_0$-sized $\gamma$-shattered set $S$. We take $\delta < 1/15$ and define the distribution $\mathcal{D}$ on $S$ as follows.

$$\mathbb{P}[\mathbf{z}_1] = 1 - 16\epsilon, \quad \mathbb{P}[\mathbf{z}_i] = \frac{16\epsilon}{d-1} \; \forall i \in \{2, \dots, d-1\}.$$

We draw a set $X$ of $m = \frac{d-1}{64\epsilon}$ i.i.d. samples from this distribution. We let $A$ be an algorithm that takes $X$ as the input and then produces a mapping $A(X) : \mathcal{Z} \to \Delta(\mathcal{A})$. We define

$$\mathrm{er}_{\mathcal{D},\gamma,h}(A(X)) := \mathbb{P}_{\mathbf{z}\sim\mathcal{D}}[r(\mathbf{z}, A(X)(\mathbf{z}), f^{(h(\mathbf{z}))}) > \gamma].$$

It suffices to establish that there exists $h^* \in \mathcal{H}$ such that $\mathbb{P}_{X\sim\mathcal{D}^m}\left[\mathrm{er}_{\mathcal{D},\gamma,h^*}(A(X)) > \epsilon\right] > 1/15$. First consider the event

$$B : |\{\mathbf{x} \in X : \mathbf{x} \in \{\mathbf{z}_2, \dots, \mathbf{z}_n\}\}| \leq (d-1)/2.$$

Using Markov's inequality, it is easy to show that $\mathbb{P}[B] \geq 1/2$. Let $U = \{\mathbf{z}_2, \ldots, \mathbf{z}_n\} \setminus X$ be the points excluding $\mathbf{z}_1$ that are unseen by the algorithm through the training set. Given $B$ we know that $|U| \geq \frac{d-1}{2}$. For the remainder of this section we will focus on

$$\mathrm{er}'_{\mathcal{D},\gamma,h}(A(X)) := \mathbb{P}_{\mathbf{z} \sim \mathcal{D}}[r(\mathbf{z}, A(X)(\mathbf{z}), f^{(h(\mathbf{z}))})$$
$$> \gamma \wedge \mathbf{z} \in \{\mathbf{z}_2, \ldots, \mathbf{z}_{n_0}\}],$$

which lower bounds $\mathrm{er}_{\mathcal{D},\gamma,h}(A(X))$. Also, we consider the following way to choose a true hypothesis $h$. First, we choose vector $b$ uniformly at random from the set $\{0,1\}^{n_0}$. We then set $h(\mathbf{z}_i) = g_0(\mathbf{z}_i)$ for all $b_i = 0$ and $h(\mathbf{z}_i) = g_1(\mathbf{z}_i)$ for all $b_i = 1$. We note that choosing a random $h$ in this way is equivalent to flipping a fair coin for each point in $S$ to determine its label. Since for any point $\mathbf{z}_i \in U$ both labels $g_0(\mathbf{z}_i)$ and $g_1(\mathbf{z}_i)$ are consistent with the training set $X$ (by definition of an SN-shattered set), we have that $A(X)$ is independent of the labeling of points in $U$. We also know that if we condition on the point drawn from $\mathcal{D}$ being some unseen point $\mathbf{u} \in U$, we get that for any strategy $A(X)(\mathbf{u}) \in \Delta(\mathcal{A})$ of the learner $\mathrm{er}'_{\mathcal{D},\gamma,h}(A(X)) \geq 1$ if $h(\mathbf{u}) = \arg\max_{j \in \{g_0(\mathbf{u}), g_1(\mathbf{u})\}} r(\mathbf{u}, A(X)(\mathbf{u}), f^{(j)})$ and 0 otherwise.

Since we are randomly choosing $h(\mathbf{z})$ to be $g_0(\mathbf{z}_i)$ or $g_1(\mathbf{z}_i)$, we have that the two conditions above happen with probability $1/2$ each. Given that there are at least $(d-1)/2$ unseen points, each sampled with probability $\frac{16\epsilon}{d-1}$, and each forcing the learner to make a $\gamma$-error with probability $1/2$, we get that

$$\mathbb{E}_{h,X}\left[\mathrm{er}'_{\mathcal{D},\gamma,h}(A(X)) \mid B\right] > \frac{1}{2} \cdot \frac{16\epsilon}{d-1} \cdot \frac{(d-1)}{2} = 4\epsilon.$$

From this analysis and the probability of $B$, we get that

$$\mathbb{E}_{h,X}\left[\mathrm{er}'_{\mathcal{D},\gamma,h}(A(X))\right] = \mathbb{E}_{h,X}\left[\mathrm{er}'_{\mathcal{D},\gamma,h}(A(X)) \mid B\right] \cdot \mathbb{P}[B]$$
$$> 4\epsilon \cdot \frac{1}{2} = 2\epsilon.$$

So there must exists some $h^*$ such that $\mathbb{E}_X\left[\mathrm{er}'_{\mathcal{D},\gamma,h^*}(A(X))\right] > 2\epsilon$. We take $h^*$ as the target concept and show that $A$ is likely to produce high error rate. We know that

$$\mathbb{P}_X\left[\mathrm{er}_{\mathcal{D},\gamma,h^*}(A(X)) > \epsilon\right] \cdot \mathbb{E}_X\left[\mathrm{er}_{\mathcal{D},\gamma,h^*}(A(X)) \mid \mathrm{er}_{\mathcal{D},\gamma,h}(A(X)) > \epsilon\right]$$
$$+ (1 - \mathbb{P}_X\left[\mathrm{er}_{\mathcal{D},\gamma,h^*}(A(X)) > \epsilon\right]) \cdot \mathbb{E}_X\left[\mathrm{er}_{\mathcal{D},\gamma,h^*}(A(X)) \mid \mathrm{er}_{\mathcal{D},\gamma,h^*}(A(X)) \leq \epsilon\right] > 2\epsilon$$

Given that for any algorithm, $A$, it holds that $\mathrm{er}'_{\mathcal{D},\gamma,h}(A(X)) \leq \mathbb{P}[\mathbf{z} \in \{\mathbf{z}_2, \ldots, \mathbf{z}_n\}] = 16\epsilon$ we get

$$\mathbb{E}_X\left[\mathrm{er}'_{\mathcal{D},\gamma,h}(A(X)) \mid \mathrm{er}_{\mathcal{D},\gamma,h}(A(X)) > \epsilon\right] \leq 16\epsilon \quad \text{for any } h$$

which implies that for $h^*$

$$\mathbb{P}_X\left[\mathrm{er}'_{\mathcal{D},\gamma,h^*}(A(X)) > \epsilon\right] \cdot 16\epsilon + (1 - \mathbb{P}_X\left[\mathrm{er}'_{\mathcal{D},\gamma,h^*}(A(X)) > \epsilon\right]) \cdot \epsilon > 2\epsilon$$
$$\iff \mathbb{P}_X\left[\mathrm{er}'_{\mathcal{D},\gamma,h^*}(A(X)) > \epsilon\right] > 1/15.$$

Therefore, the learner needs at least $m_A(\mathcal{H}, \mathcal{G}; \epsilon, \delta, \gamma) \geq \frac{d-1}{64\epsilon}$ samples.

We now consider the case, where $X$ contains only $\mathbf{z}_1$. This occurs with probability $(1 - 16\epsilon)^m \geq e^{-32m\epsilon}$, which is greater than $\delta$, when $m \leq \frac{\log(1/\delta)}{32\epsilon}$. Therefore we get that for any algorithm $A$:

$$m_A(\mathcal{H}, \mathcal{G}; \epsilon, \delta, \gamma) \geq \max\left\{\frac{d-1}{64\epsilon}, \frac{\log(1/\delta)}{16\epsilon}\right\} = C_0 \frac{\mathrm{SNdim}_{\mathcal{G}}^{(\gamma)}(\mathcal{H}) + \log(\frac{1}{\delta})}{\epsilon}$$

Lastly, we consider the case, where $\mathrm{SNdim}_{\mathcal{G}}^{(\gamma)}(\mathcal{H}) = \infty$. Then for any shattered set of size $n \in \mathbb{N}$, we can repeat the argument above, which would show that the cut-off sample complexity of any algorithm is at least $\Omega\left(\frac{n + \log(\frac{1}{\delta})}{\epsilon}\right)$, thus establishing that $m(\mathcal{H}, \mathcal{G}; \epsilon, \delta, \gamma) = \infty$. $\qquad\square$

**Theorem 4.7.** *For any class $\mathcal{H}$, game, $\mathcal{G}$, and $\epsilon, \delta, \gamma \in (0,1)$, there is some constant $C_1$ such that Algorithm 2 achieves*

$$m_{\mathfrak{L}^*}^r(\mathcal{H}, \epsilon, \delta, \gamma) \leq C_1 \frac{\mathrm{SGdim}_{\mathcal{G}}^{(\gamma)}(\mathcal{H}) \log(1/\epsilon) + \log(1/\delta)}{\epsilon}.$$

*Proof.* We work with the cut-off loss problem for some parameters $(\epsilon, \delta, \gamma) \in (0,1)^3$. We define the cut-off loss

$$\mathrm{er}_{\mathcal{D}, \gamma}(h) := \mathbb{P}_{\mathbf{z} \sim \mathcal{D}} \left[ \inf_{\mathbf{x} \in \Delta(\mathcal{A})} \max_{j \in \{h(\mathbf{z}), h^*(\mathbf{z})\}} r(\mathbf{z}, \mathbf{x}, f^{(j)}) > \gamma \right]$$

and the empirical cut-off loss for a dataset $S \in (\mathcal{Z} \times [K])^n$

$$\hat{\mathrm{er}}_{S, \gamma}(h) := \frac{1}{n} \sum_{(\mathbf{z}_i, f_i) \in S} \mathbb{1} \left\{ \inf_{\mathbf{x} \in \Delta(\mathcal{A})} \max_{j \in \{h(\mathbf{z}), h^*(\mathbf{z})\}} r(\mathbf{z}, \mathbf{x}, f^{(j)}) > \gamma \right\}$$

and the empirical loss

$$\hat{\mathrm{er}}_S(h) := \frac{1}{n} \sum_{(\mathbf{z}_i, f_i) \in S} \mathbb{1} \left\{ h(\mathbf{z}_i) \neq h'(\mathbf{z}_i) \right\}.$$

For $\mathcal{M} = \mathcal{Z} \times [K]$ we are interested in the following set

$$Q_{\epsilon, \gamma} = \{ M \in \mathcal{M}^n : \exists h \in \mathcal{H} : \hat{\mathrm{er}}_M(h) = 0, \mathrm{er}_{\mathcal{D}, \gamma}(h) > \epsilon \}$$

Now using a classic symmetrization argument (Shalev-Shwartz & Ben-David, 2014; Attias et al., 2023), we can upper bound the probability of $Q_{\epsilon, \gamma}$ with that of the following set

$$R_{\epsilon, \gamma} = \{ (N, S) \in \mathcal{M}^n \times \mathcal{M}^n : \exists h \in \mathcal{H} : \hat{\mathrm{er}}_N(h) = 0, \hat{\mathrm{er}}_{S, \gamma}(h) \geq \epsilon/2 \}.$$

The idea here is to bound the probability that there exists a hypothesis yielding maximum utility for the leader over the sample, but is $\gamma$-far from the optimal hypothesis, by the probability that this hypothesis has poor performance on another sample. More concretely, it can be shown that $\mathcal{D}^n(Q_{\epsilon, \gamma}) \leq 2\mathcal{D}^{2n}(R_{\epsilon, \gamma})$ for $n \geq c/\epsilon$, for some constant $c$. To see this, we first write

$$\mathcal{D}^{2n}(R_{\epsilon, \gamma}) = \int_{Q_{\epsilon, \gamma}} \mathcal{D}^n(\{ S \in \mathcal{M}^n : \exists h \in \mathcal{H} : \hat{\mathrm{er}}_N(h) = 0, \hat{\mathrm{er}}_{S, \gamma}(h) \geq \epsilon/2 \}) d\mathcal{D}^n(N)$$

For some $N \in Q_{\epsilon, \gamma}$ consider some $h_N \in \mathcal{H}$ that satisfies $\hat{\mathrm{er}}_{N, \gamma}(h_N) = 0$ and $\mathrm{er}_{\mathcal{D}, \gamma}(h_N) > \epsilon$. We show that $\mathcal{D}(\hat{\mathrm{er}}_{S, \gamma}(h_S) \geq \epsilon/2) \geq 1/2$, from which the claim follows straightforwardly. We know that $\mathrm{er}_{\mathcal{D}, \gamma}(h_N) > \epsilon$, and therefore $n \cdot \hat{\mathrm{er}}_{S, \gamma}(h_N) \leq n \cdot \hat{\mathrm{er}}_S(h_N)$ follows a binomial distribution with probability of success at least $\epsilon$. We note that the inequality above follows since mispredicting the follower type is a necessary condition for suboptimal utility. Using a Chernoff bound, we can show that

$$\mathcal{D}^n(S : \hat{\mathrm{er}}_{S, \gamma}(h_N) < \epsilon/2) \leq e^{-n\epsilon/8}$$

and for $n = c/\epsilon$, we get the desired result.

The next step is to use a random swap argument to upper bound $\mathcal{D}^{2n}(R_{\epsilon, \gamma})$ with something that is independent of $\mathcal{D}$. We let $\Gamma_n$ denote the set of all permutations on $\{1, \ldots, 2n\}$ that potentially swap the $i$'th and $n + i$'th elements for all $i \in [n]$. So for all $\sigma \in \Gamma_n$ and $i \in [n]$, we have that either $\sigma(i) = i, \sigma(n+i) = n+i$ or $\sigma(i) = n+i, \sigma(n+i) = i$. From Lemma 3 in Attias et al. (2023) (cf. also (Anthony & Bartlett, 1999)), we get that for some $\mathcal{M} = \mathcal{Z} \times [K]$ and distribution $\mathcal{D}$ on $V$:

$$\mathcal{D}^{2n}(R_{\epsilon, \gamma}) = \mathbb{E}_{Z \sim \mathcal{D}^{2n}} \mathbb{P}_{\sigma \sim \mathrm{Unif}(\Gamma_n)}[\sigma(V) \in R_{\epsilon, \gamma}] \leq \max_{Z \in \mathcal{Z}^{2n}} \mathbb{P}_{\sigma \sim \mathrm{Unif}(\Gamma_n)}[\sigma(V) \in R_{\epsilon, \gamma}].$$

The last step of the proof, requires reducing to learning a *partial* binary hypothesis class and using a disambiguation argument to turn the partial class into a total one, as in Attias et al. (2023). Concretely, we start by fixing a set $V \in \mathcal{M}^{2n}$,

where $v_i = (\mathbf{z}_i, h^*(\mathbf{z}_i))$ and a set $S = \{\mathbf{z}_1, \ldots, \mathbf{z}_{2n}\}$. For the projection, $\mathcal{H}(S)$ of the hypothesis class on the set, one can define the partial binary hypothesis class

$$\mathcal{H} := \left\{ h' \in \{0, 1, \star\}^{2n} : \exists h \in \mathcal{H}|_S : \forall i \in [2n] \;\; h'(\mathbf{z}_i) = \begin{cases} 0 & \text{if } h(\mathbf{z}_i) = h^*(\mathbf{z}_i) \\ 1 & \text{if } \inf_{\mathbf{x} \in \Delta(\mathcal{A})} \max_{j \in \{h(\mathbf{z}), h^*(\mathbf{z})\}} r(\mathbf{z}, \mathbf{x}, f^{(j)}) > \gamma \\ \star & \text{if } \inf_{\mathbf{x} \in \Delta(\mathcal{A})} \max_{j \in \{h(\mathbf{z}), h^*(\mathbf{z})\}} r(\mathbf{z}, \mathbf{x}, f^{(j)}) \leq \gamma \end{cases} \right\}$$

It is easy to verify that $\text{VCdim}(\mathcal{H}) \leq \text{SGdim}_{\mathcal{G}}^{(\gamma)}(\mathcal{H})$. We can now turn this partial concept class into a total concept class of size $n^{O(\text{SGdim}_{\mathcal{G}}^{(\gamma)}(\mathcal{H}) \log(n))}$ using the following disambiguation argument.

**Lemma C.3** ((Alon et al., 2021)). *Let $\mathcal{H}'$ be a partial concept class on a finite space $S$, with $\text{VCdim}(\mathcal{H}') = d$. Then there exists a disambiguation $\tilde{\mathcal{H}}'$ of $\mathcal{H}'$ of size $|\tilde{\mathcal{H}}'| = |S|^{O(d \log |S|)}$.*

We have that $\sigma(V) \in R_{\epsilon, \gamma}$ iff there exists some $h \in \mathcal{H}$ such that $h(\mathbf{z}_{\sigma(i)}) = h^*(\mathbf{z}_{\sigma(i)}) \;\; \forall \, i \in [n]$ and

$$\frac{1}{n} \sum_{i=1}^{n} \mathbb{1} \left\{ \inf_{\mathbf{x} \in \Delta(\mathcal{A})} \max_{j \in \{h(\mathbf{z}_{\sigma(n+i)}), h^*(\mathbf{z}_{\sigma(n+i)})\}} r(\mathbf{z}_{\sigma(n+i)}, \mathbf{x}, f^{(j)}) > \gamma \right\} \geq \epsilon/2.$$

We know that if there exists such an $h$ that is correct on the first $n$ points but $\gamma$-far on at least $\epsilon n/2$ of the last $n$ points, there exists a function in $\tilde{\mathcal{H}}'$, that is 0 on the first $n$ points and 1 on the points on which $h$ is $\gamma$-far. We also know that if for some $\sigma \in \Gamma_n$, $\tilde{h}' \in \tilde{\mathcal{H}}'$ is a witness that $\sigma(V) \in R_{\epsilon, \gamma}$, then at least one of $\tilde{h}'(i), \tilde{h}'(i)$ must be 0, and at least $n\epsilon/2$ elements must be non-zero. Thus for a permutation drawn uniformly at random, the probability that all 1's are on the last n elements is at most $2^{-n\epsilon/2}$. Also since $|\tilde{\mathcal{H}}'|$ is bounded we an use a union bound that yields $\mathbb{P}_{\sigma \sim \text{Unif}(\Gamma_n)}[\sigma(V) \in R_{\epsilon, \gamma}] \leq n^{O(\text{SGdim}_{\mathcal{G}}^{(\gamma)}(\mathcal{H}) \log(n))} 2^{-n\epsilon/2}$. Given that $V \in \mathcal{M}^{2n}$, we can take a $\max$ over all $V \in \mathcal{M}^{2n}$, which gives the desired result. $\qquad \square$

### C.1. Learning without contexts

Suppose there is a fixed, unknown distribution, $\mathcal{D}$, over the set of possible follower types. We denote with $u(\mathbf{x}, b_{f^{(i)}}(\mathbf{x}))$ the leader's payoff when committing to a (possibly mixed) strategy $\mathbf{x} \in \Delta(\mathcal{A})$ against a follower of type $f^{(i)}$. We use the notation $u_{\mathcal{D}}(\mathbf{x}) = \mathbb{E}_{f \sim \mathcal{D}}[u(\mathbf{x}, b_f(\mathbf{x}))]$ and given a sample $S = \{f_1, \ldots, f_m\}$ of follower types, we write $u_S(\mathbf{x}) = \frac{1}{m} \sum_{f \in S} u(\mathbf{x}, b_f(\mathbf{x}))$.

Given such a sample, the goal of the leader is to choose a mixed strategy that fares well in expectation against any new sample from $\mathcal{D}$. Concretely, we want to choose $\hat{\mathbf{x}} \in \Delta(\mathcal{A})$ such that

$$\mathbb{P}_S \left[ \sup_{\mathbf{x}} u_{\mathcal{D}}(\mathbf{x}) - u_{\mathcal{D}}(\hat{\mathbf{x}}) > \epsilon \right] \leq \delta.$$

To achieve this, it suffices to show a generalization guarantee *i.e.*, a bound on the difference between the leader's expected payoff and the empirical payoff over a set of follower types for any possible mixed strategy.

**Definition C.4** (Generalization guarantee). A generalization guarantee for the leader in a Stackelberg game, $\mathcal{G}$, is a function $\epsilon_{\mathcal{G}} : \mathbb{Z}_{\geq 1} \times (0, 1) \to \mathbb{R}_{\geq 0}$ such that for any $\delta \in (0, 1)$, any $m \in \mathbb{Z}_{\geq 1}$, and any distribution $\mathcal{D}$ over the follower types, with probability at least $1 - \delta$ over the draw of a set $S \sim \mathcal{D}^m$ for any $\mathbf{x} \in \Delta(\mathcal{A})$, the difference between the average payoff of $\mathbf{x}$ over $S$ and the expected payoff of $\mathbf{x}$ over $\mathcal{D}$ is at most $\epsilon_{\mathcal{G}}(m, \delta)$:

$$\mathbb{P}_{S \sim \mathcal{D}^m} \left[ \exists \mathbf{x} \in \Delta(\mathcal{A}) \;\; \text{s.t.} \;\; |u_{\mathcal{D}}(\mathbf{x}) - u_S(\mathbf{x})| > \epsilon_{\mathcal{G}}(m, \delta) \right] \leq \delta$$

This implies a relationship between the utility of the leader's strategy maximizing the average payoff over the training data to the expected utility of the optimal strategy. Formally, for $\hat{\mathbf{x}} \in \arg\sup_{\mathbf{x} \in \Delta(\mathcal{A})} u_S(\mathbf{x})$ and $\mathbf{x}^* \in \arg\sup_{\mathbf{x} \in \Delta(\mathcal{A})} u_{\mathcal{D}}(\mathbf{x})$

$$\mathbb{P}_{S \sim \mathcal{D}^m} \left[ |u_{\mathcal{D}}(\mathbf{x}^*) - u_S(\hat{\mathbf{x}})| > 2\epsilon_{\mathcal{G}}(m, \delta) \right] \leq \delta.$$

We let $\mathcal{P} = \{u_{\mathbf{x}} : [K] \to \mathbb{R} \text{ such that } u_{\mathbf{x}}(f) = u(\mathbf{x}, b_f(\mathbf{x}))\}$ be the set of payoff functions for the leader. Each payoff function is parameterized by the leader's mixed strategy, takes as input the follower's type, and outputs the leader's utility. We also define the dual class $\tilde{\mathcal{P}} = \{\tilde{u}_f : \Delta(\mathcal{A}) \to \mathbb{R} \text{ such that } \tilde{u}_f(\mathbf{x}) = u(\mathbf{x}, b_f(\mathbf{x}))\}$. Consider the following definition, introduced by Balcan et al. (2018b) and adapted to our setting.

**Definition C.5.** A class $\mathcal{H}$ of hypotheses, $h : [K] \to \mathbb{R}$, is $(d,t)$-*delineable* if

- The class $\mathcal{H}$ consists of hypotheses that are parameterized by vectors $\mathbf{x}$ from a set $\mathcal{X} \subseteq \mathbb{R}^d$,

- For any $f \in [K]$, there is a set $\mathcal{J}$ of $t$ hyperplanes such that for any connected component $\mathcal{X}' = \mathcal{X} \setminus \mathcal{J}$, the dual function $h_f(\mathbf{x})$ is linear over $\mathcal{X}'$.

We can now show that the primal class $\mathcal{P}$ is deliniable, which is a property we will use to show a generalization guarantee.

**Lemma C.6.** *The class of leader payoff functions, $\mathcal{P}$ that are parameterized by the leader's mixed strategy and take as input the follower's type is $(|\mathcal{A}|, \binom{|\mathcal{A}_f|}{2})$-delineable.*

*Proof.* The statement holds from the following two facts. First, given a fixed follower type, $f$, we have that $\tilde{u}_f(\mathbf{x})$ is a piecewise linear function. Second, for any fixed follower type, $f$, we have that $u_f(\mathbf{x})$ will have at most $|A_f|(|A_f| - 1)$ hyperplanes going through it. Each hyperplane corresponds to a change in the follower's best-response strategies, from one pure action to another. Therefore, there can be at most $\binom{|\mathcal{A}_f|}{2}$ hyperplanes. $\qquad\square$

Next, we present the definition of *pseudo-dimension* (Pollard, 2012), which is complexity measure that characterizes a hypothesis class and in our setting helps us reason about how difficult the learning task of the leader is. Informally speaking, the more complex the hypothesis class, $\mathcal{P}$, the harder it is for the leader to distinguish between strategies, $\mathbf{x}$, which parameterize the class, and therefore, more samples are needed to learn.

**Definition C.7** (Pseudo-dimension). Let $\mathcal{F}$ be a class of functions $f : A \to \mathbb{R}$, with an abstract domain, $A$. We say that $y^{(1)}, \ldots, y^{(n)}$, *witness the shattering* of $S = \{x^{(1)}, \ldots, x^{(n)}\} \subseteq A$ if for all $T \subseteq S$, there exists a function $f_T \in \mathcal{F}$, such that for all $x^{(i)} \in T$, $f_T(x^{(i)}) \leq y^{(i)}$ and for all $x^{(i)} \notin T$, $f_T(x^{(i)}) > y^{(i)}$. We then say that the *pseudo-dimension* of $\mathcal{F}$, denoted $\mathrm{Pdim}(\mathcal{F})$, is the size of the largest set that can be shattered by $\mathcal{F}$.

Using Lemma C.6 and results from the data-driven algorithm design literature (Balcan, 2021; Balcan et al., 2024; 2025), we immediately get the following bound on the pseudo-dimension.

**Theorem C.8** (Adaptation of Lemma 3.10 in Balcan et al. (2018b)). $\mathrm{Pdim}(\mathcal{P}) \leq 9|\mathcal{A}| \log\left(4|\mathcal{A}||\mathcal{A}_f|^2\right)$.

It is known that the finiteness of the pseudo-dimension of a function class is a sufficient condition to show a generalization guarantee for the class (Pollard, 2012; Dudley, 2010). In our case, the above bound implies a generalization guarantee with function

$$\epsilon_{\mathcal{G}}(m, \delta) = O\left(\sqrt{\frac{|\mathcal{A}| \log(|\mathcal{A}||\mathcal{A}_f|^2)}{m}} + \sqrt{\frac{\ln(1/\delta)}{m}}\right).$$

**Discussion.** We note that the above generalization result is particularly strong. Informally, it asserts that for any distribution over the follower types, if the leader has so far encountered a number of followers that scales linearly with the size of her action set and logarithmically with the follower's action set, it suffices for her to play a strategy that maximizes her average utility over the followers seen so far. This result holds even for an *infinite* number of follower types and extends prior work on learning in Stackelberg games without contextual information (Letchford et al., 2009).

*Remark* C.9. It is worth noting that even when there is only one context (i.e. $\mathcal{Z}$ is a singleton), our results in this section are incomparable with those in Section 4. This is because in this section, the goal of the leader is to learn given a deterministic mapping from contexts to follower types. So, if the contexts are the same between all examples in the training set, the follower types corresponding to these examples will also be the same. In contrast, the context-free setting we consider in Appendix C.1 allows for a distribution over follower types. Thus the two bounds are generally incomparable.

---

**Algorithm 3** Multiclass Decision Lists Learning Algorithm

---

1: **input:** input variable space, $\mathcal{Z}$, and output label space, $[K]$.
2: Define the set of rules $R = \{(\ell_i, k_i) \text{ such that } \ell_i \in \{x_i, \bar{x}_i\}, x_i \in \mathcal{Z}, \ k_i \in [K]\}$
3: Set $\tilde{h} = [R]$
4: **for** $t = 1, 2, 3, \ldots$ **do**
5:    Observe $\mathbf{z}_t$
6:    **for** each set $S = \{(\ell_1, k_1), \ldots\} \in \tilde{h}$ **do**
7:      **if** there exists $(\ell_i, k_i) \in S$ such that $\ell_i = \mathbf{z}[i]$ **then**
8:        Output label $\tilde{k}_t = k_i$
9:        Observe the true label $h^*(\mathbf{z})$
10:        **if** $\tilde{k}_t \neq h^*(\mathbf{z})$ **then**
11:          Set $M = \{(\ell_i, k_i) \in S \mid \ell_i = \mathbf{z}[i] \wedge k_i \neq k_t\}$
12:          Replace $S$ in $\tilde{h}$ with $S \setminus M$ followed by $M$
13:        **else**
14:          Break
15:        **end if**
16:      **else**
17:        Continue
18:      **end if**
19:    **end for**
20: **end for**

---

# D. Appendix for Section 5: Computational complexity considerations

**Theorem 5.1.** *The class of multiclass decision lists in Example 4 is efficiently online learnable.*

*Proof.* Consider a true multiclass decision list $h^* : \{0, 1\}^n \to [K]$. We show that Algorithm 3 makes at most $O(nKL)$ mistakes in the worst case, where $n = |\mathcal{Z}|$ is the number of variables, $L$ is the length of the target decision list and $K$ is the number of possible labels. We make the following three observations: (1) At every timestep in which the algorithm makes a mistake, at least one rule is moved down by one level. (2) Any rule will be moved down a level at most $L + 1$ times, where $L$ is the length of the target decision list. (3) By induction, the $i$-th rule of the target hypothesis will never be moved below the $i$-th level of $\tilde{h}$. Therefore the number of mistake that the algorithm makes is upper bounded the by $(L + 1)(2nK + 2)$, where $2nK + 2$ is the number of all possible rules. $\qquad\square$

