# OpenReview forum: "Learning in Structured Stackelberg Games"
_ICML.cc/2026/Conference — ICML 2026 spotlight_

### Official Review · Reviewer_cqcp · 2026-02-25

**Soundness:** 4
**Presentation:** 4
**Significance:** 3
**Originality:** 3
**Overall Recommendation:** 5
**Confidence:** 2

**Summary:**

The paper provides a complete characterization of learnability in Stackelberg games. The first result is to show that the Littlestone dimension can be infinite for a learnable Stackelberg game. Thus, a finite Littlestone dimension is sufficient but not necessary.
As pointed out by the authors, the reason is simple: the leader does not have to learn the follower's best responding strategy if this is irrelevant for her own utility. Motivated by this observation, the authors introduce the new notion of Littlestone-Stacklberg dimension which lower bounds the best possible regret. Such lower bound is matched by SSOA whose analysis is simple and elegant.

Moreover, the authors also study the distributional setting providing a lower bound leveraging the new concept of $\gamma$-SN dimension and an upper bound depending on the $\gamma$-SG dimension.

**Compliance With Llm Reviewing Policy:**

Affirmed.

**Final Justification:**

The authors' response clarified my doubts, and I am therefore recommending acceptance of this work in an even more convinced manner.

**Key Questions For Authors:**

1) I suspect there are some typos in Figures 1 and 3. I think that in Figure 1 the second leaf from the left should be labeled as $h_{(2,3,1)}$ and the third from the left as $h_{(3,1,2)}$ . Similarly, in Figure 3, the first leaf from the left should be labeled as $h_{(3,1,2)}$ and the third one from the left as $h_{(2,3,1)}$. If I am wrong and you are correct, please explain why. Thank you in advance.

2) What do you mean by the notation $z_{s \leq d}$ in the definition of the the SL shattered trees ? Is there a difference between writing $z_{s \leq d}$ and $z_{s}$. Please explain it also in the main text.

3) In the distributional case, the upper and lower bounds depend on different dimensions. How big is the gap between the $\gamma$-SN and $\gamma$-SG dimension? In which cases do they coincide ? Please add this discussion in the main text too.

Minor: please make H calligraphic at the end of page 3. Morever, in the proof of Theorem 3.9 the first utility misses a closing bracket (after closing the bracket of the arguments of $b_t$ there should be a second closed bracket).

**Limitations:**

Nothing worth highlighting. Please see the questions section.

**Strengths And Weaknesses:**

The paper is nicely written, it has a clear motivation and the intuition behing the results is greatly conveyed to the reader: the leader does not have to learn the follower's best response if that does not affect the leader's utility. This observation allows to bypass the dependence on the Littlestone dimension.

---

> ### Author Rebuttal · Authors · 2026-03-28
>
> Thank you for your positive review and for pointing out several typos. We’ll be sure to fix all of these things in the next revision of the paper.
>
> > *I suspect there are some typos in Figures 1 and 3. I think that in Figure 1 the second leaf from the left should be labeled as $h_{(2,3,1)}$ and the third from the left as  $h_{(3,1,2)}$. Similarly, in Figure 3, the first leaf from the left should be labeled as $h_{(3,1,2)}$ and the third one from the left as $h_{(2,3,1)}$. If I am wrong and you are correct, please explain why. Thank you in advance.*
>
> h_{(a,b,c)} denotes the hypothesis with h(1)=f^{(a)}, h(2)=f^{(b)}, and h(3)=f^{(c)}. We will update the paper; we apologize for the somewhat confusing notation and hope that this clears things up.
>
> > *What do you mean by the notation $z_{s \leq d}$ in the definition of the the SL shattered trees ? Is there a difference between writing $z_{s \leq d}$ and $z_{s}$.*
>
> By z_{s \leq d}, we mean “a context z_s with depth at most d”. We agree that this was not stated cleanly in our submission and we will make sure to clarify in the revision.
>
> > *In the distributional case, the upper and lower bounds depend on different dimensions. How big is the gap between the $\gamma$-SN and $\gamma$-SG dimension? In which cases do they coincide ?*
>
> This situation is analogous to the relationship between Natarajan and Graph dimensions in multiclass learning: the lower bound is governed by a shattering-type notion (SG), while the upper bound is governed by a more permissive notion (SN) that captures uniform convergence-type behavior. In the classical multiclass setting, the two differ by at most O(log(# labels)). We suspect that in our setting, SN and SG will differ by at most O(log(# follower types)), although proving this is an important direction for future work.
>
> > *Minor: please make H calligraphic at the end of page 3. Morever, in the proof of Theorem 3.9 the first utility misses a closing bracket (after closing the bracket of the arguments of there should be a second closed bracket).*
>
> Thanks for pointing these things out!

---

> > ### Author Rebuttal · Reviewer_cqcp · 2026-03-31
> >
> > Thank you for the clarifications! I do not have additional questions and I keep my very positive evaluation. Please explain the $h_{(a,b,c)}$ notation that I now understand in the revision.
> >
> > Best,
> > reviewer

---

### Official Review · Reviewer_cuLQ · 2026-03-11

**Soundness:** 3
**Presentation:** 4
**Significance:** 3
**Originality:** 3
**Overall Recommendation:** 5
**Confidence:** 3

**Summary:**

This paper considers a structured version of contextual Stackelberg games: contextual information (side information) is predictive of the follower's unknown type through some mapping belonging to a hypothesis class. The paper explores learnability (utility maximizing leader mixed action) in this setting, observing from prior works that that without this structure, online learning is not really feasible (sub-linear regret not possible).

The authors show that the standard multiclass Littlestone dimension — the classic measure of online learning complexity — can be arbitrarily loose for this problem since it considers predicting types whereas the goal here is utility oriented. In other words, mispredicting the follower's type doesn't necessarily hurt the leader's utility. They introduce the Stackelberg-Littlestone (SL) dimension, which jointly captures the complexity of the hypothesis class and the game's utility structure. They prove it tightly characterizes the leader's optimal regret (matching upper and lower bounds), and provide an optimal algorithm (SSOA) that achieves this bound.

They also consider online learning in the distributional setting. PAC learning guaratees are given with respect to two new dimensions — the $\gamma$-SN dimension (lower bound) and $\gamma$-SG dimension (upper bound). These adapt ideas from Natarajan and graph dimensions to account for the fact that only follower-type disagreements that actually affect the leader's utility matter.

**Compliance With Llm Reviewing Policy:**

Affirmed.

**Key Questions For Authors:**

See above.

**Limitations:**

I don't see any negative implications of this work.

**Strengths And Weaknesses:**

I am generally quite positive about this paper. It takes a meaningful problem (contextual Stackelberg games) and considers a structured version of this game where better learning guarantees can be given. I found the resulting analysis therein quite thorough and insightful. The fact that optimal strategy depends on the utility and this makes learning easier is quite clean. [1] also uses a similar observation to improve learnability in bayesian Stackelberg games. The bounds in the adversarial setting are tight. Further, the authors also do a reasonable analysis in the distributional setting.

I think it would improve the paper if the authors could comment on scenarios where this mapping between context and types is relevant or appropriate. The work includes an initial discussion, but more could be said (perhaps in the appendix). Regarding the computational complexity of determining the optimal strategy given a fixed context and type, I would also appreciate a bit more discussion. I think it's an important problem that was (perhaps too easily) side-stepped here. The classic result of [Conitzer & Sandholm, 2006] is indeed negative, but I am curious if you think more recent works probing at some positive directions [1] are useful here? Co-incidentally, this work also uses the utility trick to decrease learning

[1]: Learning to Play Multi-Follower Bayesian Stackelberg Games; Gerson Personnat, Tao Lin, Safwan Hossain, David C. Parkes; ICLR 2026.

---

> ### Author Rebuttal · Authors · 2026-03-28
>
> Thanks for your positive review and for taking the time to review our submission and for pointing out reference [1]; we will include it in the final version.
>
> > *I think it would improve the paper if the authors could comment on scenarios where this mapping between context and types is relevant or appropriate. The work includes an initial discussion, but more could be said (perhaps in the appendix).*
>
> We’d be happy to include such a discussion in the appendix. In short, our methods are relevant in Stackelberg games where the side information says something about the follower’s type, not just the utility of the respective players. As an example, we’d argue that this captures several security game settings: in wildlife protection, specific tire tracks might signal the presence of a particular poaching group and market demand trends might be indicative of which animals’ poachers are now more likely to appear. Similarly, in airport security, current affairs often raise or lower the risk of an attack on certain international flights.
>
> We’d also like to point out our discussion in Appendix A, where we highlight how our results are additionally applicable to the related problems of (1) simultaneous second-price auctions with side information and (2) Bayesian persuasion with both public and private states.
>
> > *Regarding the computational complexity of determining the optimal strategy given a fixed context and type, I would also appreciate a bit more discussion. I think it's an important problem that was (perhaps too easily) side-stepped here. The classic result of [Conitzer & Sandholm, 2006] is indeed negative, but I am curious if you think more recent works probing at some positive directions [1] are useful here?*
>
> Thanks for raising this point. Our work is primarily focused on the statistical learnability of the problem rather than computational efficiency. As in the classical online learning literature (e.g. the Standard Optimal Algorithm for Littlestone dimension), our algorithms should be viewed as information-theoretic constructions, and may not be computationally efficient for general hypothesis classes.
>
> With that being said, we view fleshing out the computational considerations of learning in our setting as an important direction for future research. As a starting point, we have included a preliminary discussion on efficient learning algorithms in Appendix D.
>
> As you point out, [1] exploits a geometric best-response region decomposition under a stochastic type-distribution model, which enables efficient optimization when the number of leader actions is small. We hypothesize that one could exploit a similar decomposition in our setting with side information (for each fixed context). However this does not allow us to side-step the computation of the SL dimension, which may be computationally challenging.
>
> That said, we believe these directions are highly complementary. An interesting open question is whether one can combine our learnability guarantees with additional structural assumptions (e.g., low-dimensional or geometric structure) to obtain computationally efficient algorithms.

---

> > ### Author Rebuttal · Reviewer_cuLQ · 2026-03-31
> >
> > Thank you for your response.

---

### Official Review · Reviewer_9aqk · 2026-03-12

**Soundness:** 3
**Presentation:** 3
**Significance:** 3
**Originality:** 3
**Overall Recommendation:** 4
**Confidence:** 1

**Summary:**

This paper studies learnability in contextual Stackelberg games, where both the leader’s and the follower’s payoffs depend on additional side information. The paper first shows that the Littlestone dimension does not characterize the difficulty of maximizing the leader’s payoff in this setting. It then introduces a new complexity notion and proves that it tightly characterizes the complexity of the learning problem. Related theoretical guarantees are also established for the distributional setting.

**Compliance With Llm Reviewing Policy:**

Affirmed.

**Key Questions For Authors:**

My main concerns and questions are already included in the Weaknesses section above.

**Limitations:**

I see no negative societal impacts that need to be addressed.

**Strengths And Weaknesses:**

### Strengths

The paper is well organized, and the theoretical results appear to be stated rigorously. Although I am not deeply familiar with Littlestone dimension or this line of complexity analysis, I found the central insight interesting, namely that the Littlestone dimension is not the right notion for characterizing learnability in contextual Stackelberg games.

### Weaknesses

My main concern is about the proposed online learning algorithm and its practical meaning. In particular, it is unclear whether SSOA is intended to be computationally implementable in a realistic sense. For example, Algorithm 1 appears to require computing the SL dimension (see line 6), and it is not discussed whether this quantity can actually be computed within a reasonable running time. If not, it would be helpful for the paper to clarify whether Algorithm 1 should instead be viewed mainly as a theoretical existence result showing that the regret upper bound matching the lower bound in Theorem 3.8 is achievable in principle.

A related question concerns the assumptions required to run the algorithm. In practice, computing an exact best response may itself be difficult, and in some settings the learner may not have direct access to the payoff function $u$. It would therefore be valuable to discuss whether an online learning algorithm achieving a regret guarantee comparable to that of Theorem 3.9 can still be obtained under such weaker computational or informational assumptions.

---

> ### Author Rebuttal · Authors · 2026-03-28
>
> Thank you for your positive review and for taking the time to review our submission.
>
> > *My main concern is about the proposed online learning algorithm and its practical meaning. In particular, it is unclear whether SSOA is intended to be computationally implementable in a realistic sense. For example, Algorithm 1 appears to require computing the SL dimension (see line 6), and it is not discussed whether this quantity can actually be computed within a reasonable running time. If not, it would be helpful for the paper to clarify whether Algorithm 1 should instead be viewed mainly as a theoretical existence result showing that the regret upper bound matching the lower bound in Theorem 3.8 is achievable in principle.*
>
> You’re right that Algorithm 1 (SSOA) should primarily be viewed as an algorithm establishing that the regret upper bound in Theorem 3.8 is achievable. This mirrors the role of the Standard Optimal Algorithm for the Littlestone dimension, which is used to characterize optimal regret in online classification despite not being efficiently implementable in general. That is to be expected since for most learning problems, the algorithms that achieve optimal bounds are not computationally efficient. Yet, it is important to understand what is achievable in principle.
>
> We would also like to note that for specific hypothesis classes with additional structure, efficient implementations may be possible, and exploring this is an interesting direction for future work. We include a preliminary exploration of computationally-efficient algorithms in our setting in Appendix D.
>
> >*A related question concerns the assumptions required to run the algorithm. In practice, computing an exact best response may itself be difficult, and in some settings the learner may not have direct access to the payoff function . It would therefore be valuable to discuss whether an online learning algorithm achieving a regret guarantee comparable to that of Theorem 3.9 can still be obtained under such weaker computational or informational assumptions.*
>
> If the learner has access to a follower’s payoff function, then computing their best-response may be done in O(# follower actions) for a fixed context and leader strategy (assuming the follower breaks ties in a fixed and known way). Similarly, the optimal leader strategy to play against a given follower type can be computed by solving O(# follower actions) linear programs (Conitzer and Sandholm, 2006). Technically, our algorithms only require oracle access to the follower’s best-response function (see footnote 3). With that being said, it would be interesting to further relax this assumption, for example by only assuming access to a noisy best-response function (e.g. a function that outputs the true best response with probability p and a random action with probability 1–p). In this setting, it may be possible to get similar regret guarantees to Theorem 3.9 (with high probability, and at the cost of additional compute) by repeatedly sampling the best-response oracle at each round of Algorithm 1.
>
> We would be happy to expand upon both of these points in the final version of the paper.

---

> > ### Author Rebuttal · Reviewer_9aqk · 2026-04-03
> >
> > Thank you for your clarification. I was still interested in the possibility of extending the proposed algorithm to a bandit-feedback setting, where each player has access only to realized payoffs. However, I agree that, under such feedback, computing a best response at each iteration would be highly nontrivial. This point does not lessen my assessment of the paper’s contribution, and I am happy to maintain my positive score.

---

> > > ### Author Response · Authors · 2026-04-03
> > >
> > > Thanks for your reply! We hadn’t realized that you were asking about the bandit feedback setting.
> > >
> > > When talking about bandit feedback, one could consider two possibilities: (A) the leader observes the best-response of the follower (but not the follower’s type), or (B) the leader only observes her realized payoff (but not the follower’s type or best response).
> > >
> > > For (A), we can draw an analogy to online classification. [1] introduces an extension of the Littlestone dimension which characterizes learnability in online multiclass classification under bandit feedback, by modifying the notion of a shattered tree. In that setting, Littlestone(H) <= Bandit-Littlestone(H) <= O((# labels)*log(# labels)*Littlestone(H)). We believe that it may be possible to extend our results to this type of bandit feedback by similarly modifying our definition of shattering. In our case, we would expect to replace # labels with # follower types and Littlestone(H) with SL(H). So we do expect our learning problem here to be more challenging, probably by an amount proportional to the number of follower types. (Although this does not immediately follow from our existing results, and working this out formally is an important future direction.)
> > >
> > > For (B), the learning problem of the leader becomes more challenging, and such an extension would likely require substantially different techniques. With that being said, some very recent work (Balcan et al, 2026) is able to learn under this type of feedback in a different Stackelberg game setting, and so some of their techniques/ideas may provide a useful starting point.
> > >
> > > [1]: Multiclass Learnability and the ERM principle by Amit Daniely, Sivan Sabato, Shai Ben-David, Shai Shalev-Shwartz, 2014

---

### Official Review · Reviewer_NcZv · 2026-03-13

**Soundness:** 4
**Presentation:** 4
**Significance:** 3
**Originality:** 4
**Overall Recommendation:** 5
**Confidence:** 2

**Summary:**

The paper studies structured Stackelberg games. They are a generalization of Stackelberg games where some contextual information is revealed to both the leader and follower, and the follower has a type associated with them. They show that a measure of complexity analogous to the Littleton dimension in online classification characterises the leader's instance-optimal regret. They provide an algorithm for the leader that achieves this regret.

**Compliance With Llm Reviewing Policy:**

Affirmed.

**Final Justification:**

I recommend acceptance, I think it's a strong paper. The authors clarified the few questions I had in the review clearly.

**Key Questions For Authors:**

1) Can you explain why the realizability assumption is needed? In the related literature, you mention that this circumvents some impossibility result.
2) Without this assumption, would the notion of complexity change? Do unlearnable classes correspond to those with infinite SL dimension?

**Limitations:**

yes

**Strengths And Weaknesses:**

Strengths:
Paper is well-written and clearly states its contributions and limitations.
The result where they show that it is possible to construct a problem instance where the leader can learn to play optimally even with infinite Littlestone dimension sets the tone for why they need to define a different notion of complexity.

Weaknesses:
The realizability assumption seems a bit restrictive.

---

> ### Author Rebuttal · Authors · 2026-03-28
>
> Thank you for your positive review and for taking the time to review our submission.
>
> > *Can you explain why the realizability assumption is needed? In the related literature, you mention that this circumvents some impossibility result.*
>
> and
>
> > *Without this assumption, would the notion of complexity change? Do unlearnable classes correspond to those with infinite SL dimension?*
>
> Realizability is important in our setting for both conceptual and technical reasons. Conceptually, it is the structural assumption that allows us to overcome the impossibility result from prior work (Harris et al., 2024) for fully adversarial context/follower type sequences: rather than allowing a completely arbitrary mapping from contexts to follower types, we assume that this mapping is induced by some h^* in a known function class H. Technically, our algorithms rely on maintaining a nonempty set of hypotheses that are perfectly consistent with the observations; this is what allows our SL-based analysis to go through. In the realizable (online) setting, we show that the SL dimension is infinite if and only if the regret (with respect to h^*) scales linearly with the time horizon.
>
> In the agnostic setting (i.e. the setting where one does not assume realizability), the benchmark changes from a true h^* \in H to the best approximation in H. In agnostic online classification, there are standard reductions that allow one to get regret which scales as \Theta(\sqrt{LittlestoneDimension(H)*T}) with respect to the best hypothesis in the class. (See, e.g. “Agnostic Online Learning” by Shai Ben-David, David Pal, and Shai Shalev-Shwartz, COLT 2009.) So the notion of complexity does not change in online classification, but the guarantees one can show are weaker. We suspect that a similar reduction would apply in our Stackelberg setting (with the SL dimension in place of Littlestone), and showing this formally is an interesting direction for future research which we will add in the Discussion section of the paper.

---

> > ### Author Rebuttal · Reviewer_NcZv · 2026-04-03
> >
> > Thank you for your clarification. I'll maintain my score.

---

### Decision · Program_Chairs · 2026-04-30

**Decision:**

Accept (spotlight)

**Comment:**

The paper studies structured Stackelberg games where the leader has contextual information that can be predictive of the follower's (unknown) type. This novel game model is motivated by security games and AI safety. In the online setting, the paper shows that standard learning-theoretic measures of complexity do not characterize the difficulty of the leader's learning task. Thus, the paper introduces a novel learning-theoretic measure of complexity (termed Stackelberg-Littlestone dimension) that tightly characterizes the leader's instance-optimal regret. The paper leverages it to provide a provably optimal online learning algorithm.

All the Reviewers are positive on this paper, as it is very well written and studies a relevant problem in the literature on learning in Stackelberg games. Therefore, I strongly recommend acceptance of the paper.